# Designing strong inducible synthetic promoters in yeasts

**Masahiro Tominaga** [1,2], **Yoko Shima**[1], **Kenta Nozaki** [2], **Yoichiro Ito**[1,2], **Masataka Someda**[3], **Yuji Shoya**[3], **Noritaka Hashii** [4], **Chihiro Obata**[4], **Miho Matsumoto-Kitano**[2], **Kohei Suematsu**[1], **Tadashi Matsukawa**[1], **Keita Hosoya**[1], **Noriko Hashiba** [2], **Akihiko Kondo** [1,2,5,6] & **Jun Ishii** [1,2] ✉

Inducible promoters are essential for precise control of target gene expression in synthetic biological systems. However, engineering eukaryotic promoters is often more challenging than engineering prokaryotic promoters due to their greater mechanistic complexity. In this study, we describe a simple and reliable approach for constructing strongly inducible synthetic promoters with minimum leakiness in yeasts. The results indicate that the leakiness of yeast-inducible synthetic promoters is primarily the result of cryptic transcriptional activation of heterologous sequences that may be avoided by appropriate insulation and operator mutagenesis. Our promoter design approach has successfully generated robust, inducible promoters that achieve a $> 10^3$-fold induction in reporter gene expression. The utility of these promoters is demonstrated by using them to produce various biologics with titers up to 2 g/L, including antigens designed to raise specific antibodies against a SARS-CoV-2 omicron variant through chicken immunization.

Artificial biological activities, such as maximizing metabolite production[1], producing biotherapeutics[2,3], and reprograming cellular behavior using synthetic genetic circuits[4], may be accomplished by modulating gene expression. Inducible promoters are essential building blocks for this purpose. Specifically, decoupling the cell growth and protein production phases is a major purpose of using inducible promoters[5]. Their programmability and wide dynamic range allow for precise control of target gene expression. Consequently, identifying or engineering effective inducible promoters is a crucial step in most synthetic biology projects. Because of the lack of thorough understanding of their sequence-function relationships compared to their prokaryotic counterparts, eukaryotic inducible promoters are usually more difficult to engineer[6], even in yeast, a well-characterized eukaryotic microbial model. This can be partially attributed to their long promoter sequence containing multiple sequence motifs, in which endogenous factors bind to control promoter activity in an integrated fashion. Therefore, yeast synthetic biology still relies on a set of well-known promoters[7].

Inducible synthetic promoters (iSynPs) have been constructed to expand the yeast promoter library with desired characteristics. To enable efficient fine-tuning of inducible transcription, researchers have constructed synthetic core promoter sequences fused to various natural upstream activation sequences for ligand-responsive transcription activators[8–10]. Alternatively, synthetic transcription activators (sTAs), which involve the fusion of eukaryotic transcription activators (eTAs) to inducer-dependent DNA binding proteins, may be used to simplify yeast iSynPs[11–13], by fusing short bacterial operator sequence(s) to natural promoters lacking an upstream activating sequence (core promoters). This enables sTAs to bind upstream of the core promoter to activate downstream transcription. However, iSynPs often exhibit significant leakiness, even with a well-optimized sTA exhibiting minimal leaky transcriptional activation when deactivated.

[1]Engineering Biology Research Center, Kobe University, Kobe, Japan. [2]Graduate School of Science, Technology and Innovation, Kobe University, Kobe, Japan. [3]Pharma Foods International Co. Ltd., Kyoto, Japan. [4]Division of Biological Chemistry and Biologicals, National Institute of Health Sciences, Kawasaki, Kanagawa, Japan. [5]Department of Chemical Science and Engineering, Faculty of Engineering, Kobe University, Kobe, Japan. [6]Center for Sustainable Resource Science, RIKEN, Yokohama, Japan. ✉e-mail: junjun@port.kobe-u.ac.jp

There are at least two reasons for the sTA-independent leakiness of iSynPs. One is a transcriptional readthrough from the upstream chance promoter[4,14], and the second is a long-range activation of iSynP by endogenous transcription activators from sequences as far as 1 kbp upstream of iSynP[15–17]. Without appropriate insulation, de novo-designed iSynPs may undergo significant leakiness from one or both of these events[9,13]. Thus, the optimal design of the iSynPs remains unclear. Even with newly developed model-driven optimization technologies, designed promoters rarely outperform benchmark promoters in their induction capabilities[18].

In this study, we present a generic design for constructing tightly regulatable iSynPs in yeast (Fig. 1). We found that strong yeast iSynPs can be constructed by (1) inserting > 1-kbp insulator sequences to prevent transcriptional activation from upstream cryptic activating sequences, (2) directly fusing operators upstream of the TATA-box, and (3) increasing operator repeats and/or screening (mutating) bacterial operators to reduce their cryptic activation without compromising binding to sTAs. Based on these rules, we construct a series of inducible expression constructs with > 10^3-fold induction in reporter gene expression, one of which can be induced by removing the inducer. These induction systems are validated by the production of two different pharmaceutical proteins as well as the cost-effective, inducer-free large-scale overproduction of a single value-added protein with a titer of up to 2 g/L. We also demonstrate the utility of the promoters by producing an omicron variant of the SARS-CoV-2 receptor binding domain (RBD), which is notoriously difficult to express in microbes[19] and may be

readily used for chicken immunization to obtain antigen-specific antibodies. The expression systems described herein may be used in the future to produce various pharmaceutical proteins in a highly flexible manner for several purposes and to reprogram metabolic pathways.

## Results

### Yeast iSynP leakiness attributed to cryptic activating sequences

To design strong iSynP for yeast with minimum leakiness, we constructed a 2,4-diacetylphloroglucinol (DAPG)-iSynP as a model promoter in the methylotrophic yeast, *Komagataella phaffii* (formerly *Pichia pastoris*) (Fig. 2a). First, a 197-bp *KpAOX1* core promoter[8] was fused downstream of a single operator DNA, *phlO*. As a DAPG-regulatable sTA, rPhlTA[13] was used to bind *phlO* to activate transcription from the downstream promoter only in the presence of DAPG. This iSynP was cloned upstream of the reporter gene encoding the enhanced green fluorescent protein (EGFP) to evaluate transcription from the iSynP. The resulting expression cassette was introduced into yeast expressing rPhlTA. EGFP expression in the yeast strain was > 100-fold higher than yeast without the plasmid, even in the absence of DAPG, and was increased 8-fold following the addition of DAPG (Fig. 2b, X = 0). Note that DAPG had no detectable toxicity even at a concentration of 50 μM in *K. phaffii* (Supplementary Fig. 1).

To identify the source of the observed leakiness, we introduced a random mutation into the TATA-box sequence of the iSynP. We found that this random mutation in the TATA-box sequence decreased the induced and uninduced EGFP expression (Supplementary Fig. 4). Thus,

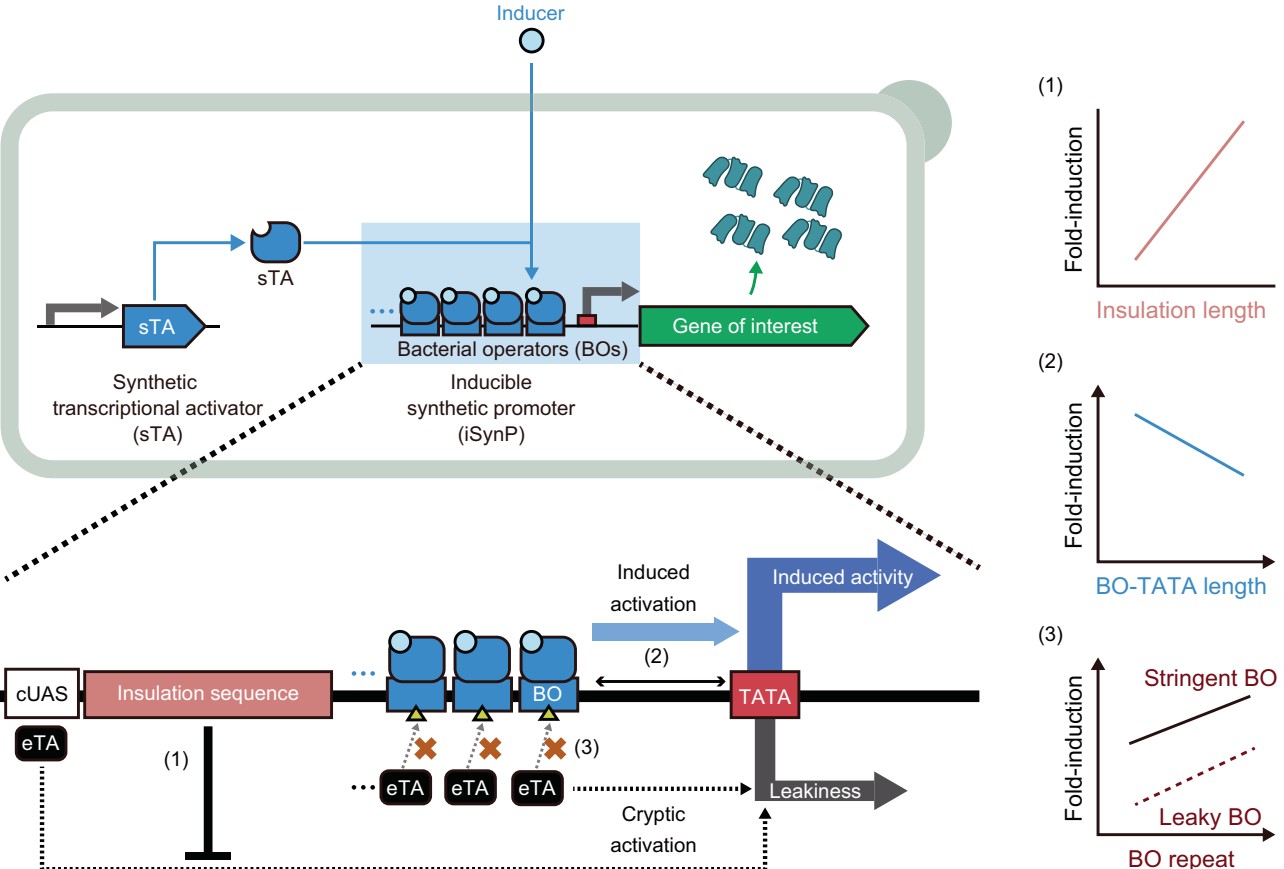

**Fig. 1 | Design strategy for high-performance inducible synthetic promoters (iSynPs) in yeasts.** iSynPs comprise two core elements: a TATA-box, and upstream bacterial operators (BOs). These operators bind to synthetic transcription activators (sTA) to induce downstream gene expression. To maximize the induction capabilities of iSynPs, we proposed the following strategies: (1) insertion of a long (> 1-kbp) insulation sequence helps prevent spurious activation by distant regulatory elements, (2) fusing the BOs directly to the TATA-box strengthens activation by sTAs, and (3) increasing BO repeats and/or introducing specific mutations to BOs can further reduce leakiness by hindering cryptic interactions with eukaryotic transcription activators (eTA). ATG, start codon; cUAS, cryptic upstream activation sequence; iSynP, inducible synthetic promoter; sTA, synthetic transcription activator; TATA, TATA-box.

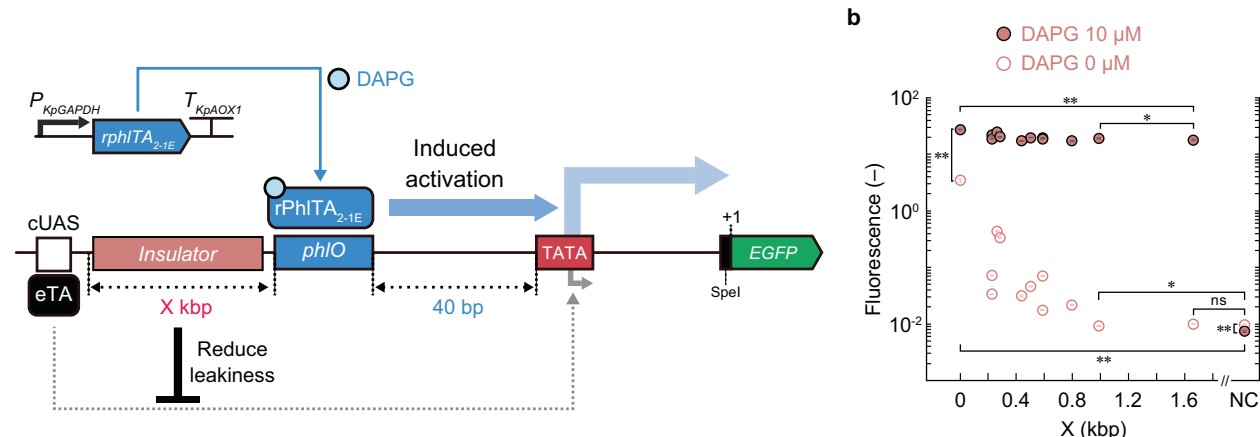

**Fig. 2 | Impact of insulation sequence on the leakiness of the DAPG-inducible synthetic promoter (iSynP) in *K. phaffii*. a** Schematic of the DAPG-iSynP system. The gene encoding rPhlTA, a fusion protein combining the PhlF transcription factor from *Pseudomonas protegens*, three copies of VP16 (VP48), and a nuclear localization signal (see details in Supplementary Fig. 2), was cloned between the *KpGAPDH* promoter ($P_{KpGAPDH}$) and *KpAOX1* terminator ($T_{KpAOX1}$) of *K. phaffii*. A 30-bp PhlF-binding sequence (*phlO*) containing a 24-bp inverted repeat sequence (as described in ref. [55]) was fused 5-bp upstream of the previously reported 197-bp *KpAOX1* core promoter[8], followed by the SpeI restriction site, *EGFP* gene, and $T_{KpAOX1}$ terminator. The absence of an insulation sequence might cause transcriptional activation of the *AOX1* core promoter by cUAS in the vector sequence due to the binding of the eTA. **b** Effect of insulation sequences on the basal activity of yeast iSynPs. The EGFP fluorescence of yeast cells containing the system with or without an insulation sequence upstream of the iSynP was determined by flow cytometry. Detailed information on insulation sequences is indicated in Supplementary Fig. 3. Error bars represent the mean ± SD of four independent experiments. The single and double asterisk represents $p < 0.05$ and $p < 0.01$, respectively. The $p$-values of the two-sided Welch's $t$ test are provided in Source Data. When comparing on- and off-state fluorescence, a paired $t$ test was used. cUAS, cryptic upstream activation sequence; DAPG, 2,4-diacetylphloroglucinol; EGFP, enhanced green fluorescent protein; eTA, eukaryotic transcriptional activator; NC, negative control (without reporter plasmid); ns, not significant; TATA, TATA-box.

we hypothesized that distal transcriptional activation rather than a transcriptional readthrough was a major source of the observed leakiness and could be avoided by inserting an appropriate insulation sequence. As expected, adding DNA fragments from *K. phaffii* with different lengths up to 1.6 kb upstream of the iSynP markedly reduced the leakiness in a length-dependent manner by up to 376-fold while minimally reducing the induced promoter activity (< 1.6-fold) (Fig. 2b). To identify the causal enhancer sequence(s) for iSynP leakiness, we randomized a 30-bp sequence just upstream of the iSynP and screened the variants with reduced leakiness (Supplementary Fig. 5). We identified 68 unique mutant sequences of which 39 sequences reduced iSynP leakiness by > 5-fold, indicating that the original 30-bp sequence may act as an "enhancer." Nonetheless, the leakiness was not completely alleviated, possibly because of the cumulative activation effect of the further upstream sequence. Taken together, these data indicate that inserting a > 1-kbp insulating sequence is a feasible way to minimize the leakiness of iSynPs (Fig. 1). In all of the subsequent experiments using *K. phaffii*, ca. 1.6-kb *KpARG4* sequence was placed upstream of iSynPs unless otherwise noted.

### Construction of strong iSynPs with minimal endogenous elements

To design iSynP with strong induced expression, we examined the effect of iSynP architecture on the induction of gene expression by spacing the length between the *phlO* and TATA-box and the start codon of the *EGFP* gene (Fig. 3a). The fold induction increased when the spacer length between the *phlO* and TATA-box decreased and almost plateaued at ≤ 40-bp for the *KpAOX1* core promoter sequence, except for the *KpDAS1* promoter with a 67-bp spacer because of the competitive binding at this position between rPhlTA and the endogenous transcription repressors. Replacing a 15-bp sequence downstream of the *phlO* element with another sequence resulted in a significantly increased fold induction of this promoter (Fig. 3b, *KpDAS1mut*). Interestingly, despite previous reports indicating that the spacing between the TATA-box and start codon can influence protein production in yeast[20], truncating the core promoters downstream had

a small effect on induction in this case (Fig. 3b); decrease in fold induction of truncation variants was at most 11% ($Y = 45$, *KpAOX1* core promoter). The resultant 94-bp iSynP with a 53-bp *KpAOX1* core promoter and the 83-bp iSynP with a 47-bp *KpDAS1* core promoter were strongly induced in the presence of DAPG, with $1731 \pm 60$- and $1148 \pm 52$-fold increases, respectively (Fig. 3d, e). Notably, the induced promoter activity of 94-bp iSynP was twofold stronger compared with that of the *KpGAPDH* promoter from *K. phaffii*. Thus, the strong minimal iSynP may only contain a bacterial operator, a TATA-box sequence, and a short downstream spacer sequence that minimally affects the induction.

To broaden the applicability of this finding, we generated four iSynPs in another yeast species, *Saccharomyces cerevisiae*, incorporating various core promoter lengths. Interestingly, an iSynP as short as 110 bp containing a 68-bp *ScGAL1* core promoter fused to *phlO*, achieved a > 100-fold induction upon DAPG exposure (Supplementary Fig. 7). This demonstrates the successful construction of potent yeast iSynPs with minimal reliance on endogenous sequences (shorter than 110 bp) in two distinct yeasts. The key to this achievement lies in the direct fusion of a bacterial operator to the TATA-box sequence.

### Repeating operators with no cryptic activation to maximize fold induction

After constructing iSynPs with minimal leakiness, we applied the design to other iSynPs using different sTAs and corresponding BOs. We designed three inducible promoters responsive to DAPG, doxycycline (Dox), and *N*-(ketocaproyl)-D,L-homoserine lactone (HSL) as follows: different copies of *phlO*, *tetO2*[21,22], and *luxO*[23], respectively, were fused upstream of the TATA-box sequence of the *KpAOX1* promoter, where rPhlTA, Dox-responsive sTA (rTetTA), and HSL-responsive sTA (LuxTA) are bound to activate transcription[13], resulting in a KpDAPG-ON, KpTet-ON, and KpHSL-ON switch, respectively (Fig. 4 and Supplementary Fig. 8). The KpDAPG-ON switch exhibited > 2000-fold induction of GFP expression upon the addition of the DAPG with negligible leakiness with 2 or more *phlO* repeats was fused to TATA-box (Fig. 4a). KpTet-ON switch exhibited > 1000-fold

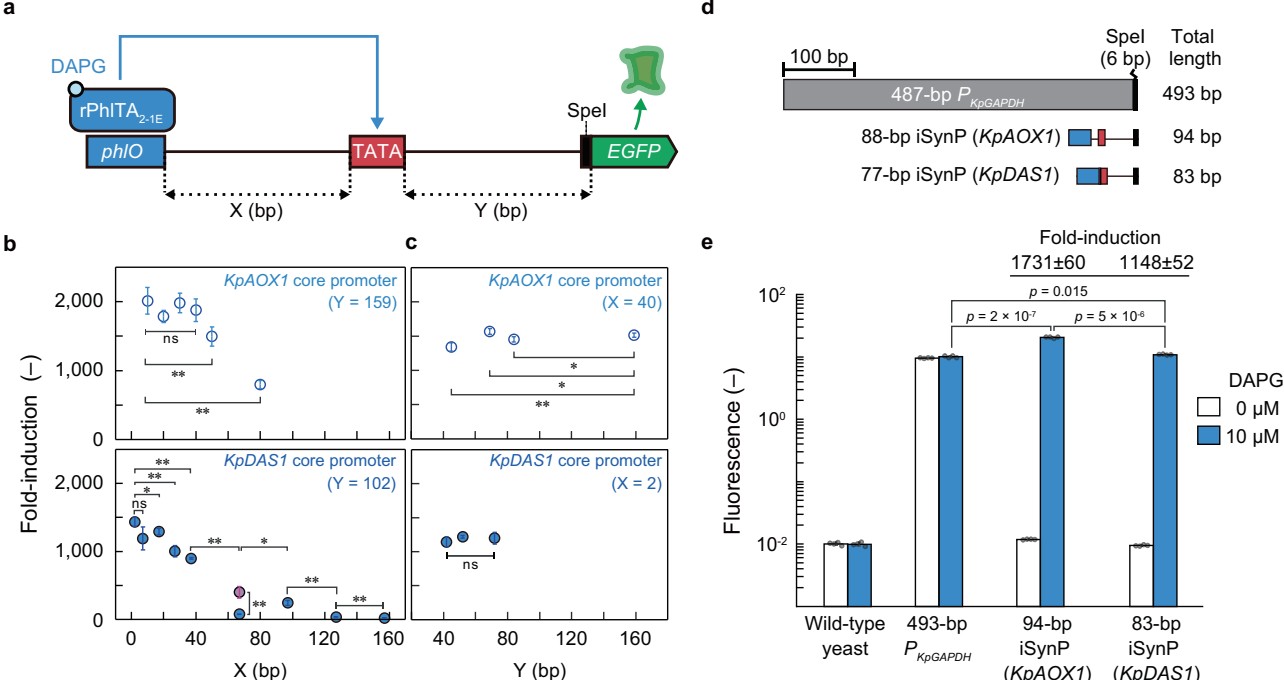

**Fig. 3 | Optimizing design constraints for strong minimal iSynPs in yeast.**
**a** Schematic illustration of DAPG-iSynPs with different spacer lengths between *phlO* element and the TATA-box (X [bp]) and between the TATA-box and the start codon (Y [bp]). **b**, **c** Effect of the sequence length between the *phlO* element and the TATA-box (**b**) and between the TATA-box and the start codon (**c**) on induced gene expression driven by the DAPG-iSynPs. Fold induction was calculated by dividing EGFP fluorescence in the presence of 10 μM DAPG by the uninduced EGFP fluorescence in its absence, as measured by flow cytometry. Magenta dot: *KpDAS1-mut*. **d**, **e** Comparison of induced gene expression level between the conventional

*KpGAPDH* promoter system ($P_{KpGAPDH}$) and two iSynPs: a 94-bp iSynP with a 53-bp *KpAOX1* core promoter and an 83-bp iSynP with a 47-bp *DAS1* core promoter. A schematic comparison of these promoters is shown in (**d**). Error bars represent the mean ± SD of four (**b** and **e**) or six (**c**) independent experiments. The single and double asterisk represents $p < 0.05$ and $p < 0.01$, respectively. The *p*-values of the two-sided Welch's *t* test are indicated and provided in Source Data. The promoter architecture of these iSynPs is shown in Supplementary Fig. 6. DAPG, 2,4-diacetylphloroglucinol; EGFP, enhanced green fluorescent protein; iSynP, inducible synthetic promoter; ns, not significant; TATA, TATA-box; WT, wild-type.

induction of GFP expression upon the addition of the Dox when 3 or more *tetO2* repeats was fused to TATA-box (Fig. 4b). Although the number of operator sequences has a nonmonotonic effect on the induction[24], increasing operator repeats monotonically increases the fold induction of the system. Notably, the leakiness of the isynPs was drastically reduced by increasing operator repeats (Supplementary Fig. 8), possibly by acting as a decoy for the eTA binding to exert cryptic activation[25,26].

The KpHSL-ON-inducible system (KpHSL-ON switch) displayed a lower induction fold (< 1000-fold) (Fig. 4c) and significant background activity without ligands even when the 10 *luxO* repeats fused to TATA-box (Supplementary Fig. 8c). We addressed this issue by replacing the part of *luxO* repeats with other *luxO* variants from *Aliivibrio fischeri* (formerly *Vibrio fischeri*)[27] to alleviate cryptic LuxTA-independent promoter activation (Supplementary Fig. 12a) while maintaining LuxTA binding. Therefore, we constructed a series of iSynPs, in which three of the five *luxO* sequences upstream of the TATA-box were replaced with three different *luxO* sequences from *A. fischeri*. As expected, one of the iSynPs with *luxO* variant from the *VF1161* promoter of *A. fischeri* ($luxO_{VF1161}$) exhibited 2-fold reduced leakiness compared to that of the original iSynP and resulted in HSL-dependent induction up to 517-fold (Supplementary Fig. 12b). Finally, with >10 *luxO* repeats including three copies of $luxO_{VF1161}$ downstream, HSL-dependent GFP induction reached >1500-fold (Fig. 4c). Consequently, a series of strong iSynPs that induced downstream reporter expression >10³-fold were constructed by fusing bacterial operator repeats directly to the TATA-box sequence and by choosing BOs that show minimal cryptic activation.

## Yeast-based biotherapeutic production platform based on the designed iSynPs

To demonstrate the capability of the promoters for inducing protein secretion, the KpDAPG-ON system was used to produce various pharmaceutical proteins, including nanobody® ALX-0171 (gontivimab), a potent treatment for respiratory syncytial virus infection[28], and nanobody® ALX-0081 (also known as caplacizumab or Cablivi®), which is used to treat acquired thrombotic thrombocytopenic purpura[29]. Genes encoding these nanobodies were C-terminally fused to the mutant secretion signal sequence, MFα (2 × Adv)[30], and cloned downstream of the 419-bp strong DAPG-iSynP, $P_{KpPhl6.0}$, containing a 6 *phlO* repeat a 61-bp spacer upstream of the TATA-box, one of the DAPG-iSynP that could be induced more than 1500-fold upon the addition of 10 μM DAPG (Fig. 5a and Supplementary Figs. 13, 14). In these strains, each protein was successfully produced only in the presence of DAPG, with titers > 5-fold higher than those obtained using a methanol induction system (*KpAOX1* promoter).

In addition to the conventional fermentation of single biologics, the rapid and single-dose production of multiple biologics with a single yeast strain at the point of care is crucial for emergencies that require production speed and flexibility rather than purity and productivity[3,31]. Therefore, we attempted to selectively produce different nanobodies with a single yeast strain. The yeast strain harboring genes encoding ALX-0171 and ALX-0081 downstream of the KpDAPG-ON and KpTet-ON systems successfully secreted ALX-0171 and ALX-0081 into the supernatant, but only in the presence of the corresponding inducer (Fig. 5b).

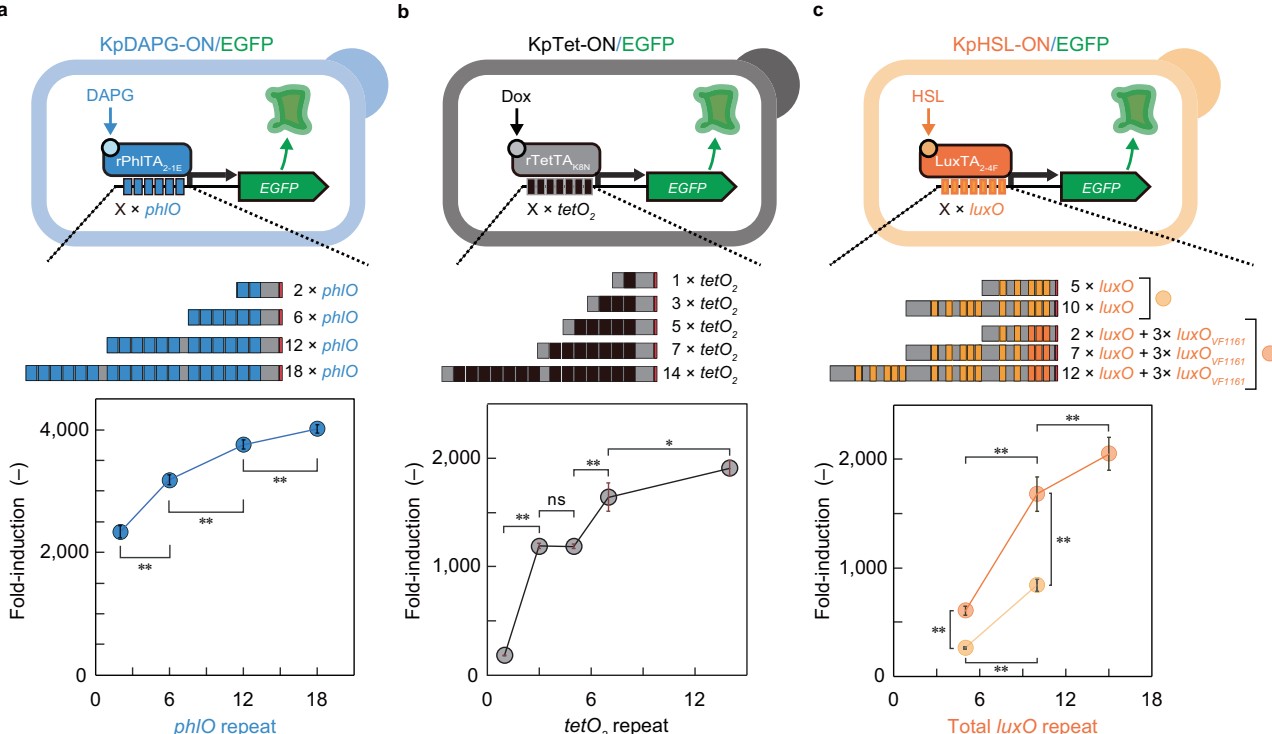

**Fig. 4 | Construction of diverse synthetic inducible switches. a–c** Schematic illustrations of iSynPs in the inducible systems responsive to DAPG (**a**), Dox (**b**), and HSL (**c**), respectively, constructed in *K. phaffii*. These systems are named KpDAPG-ON, KpTet-ON, and KpHSL-ON, respectively. The promoter architecture of these iSynPs is shown in Supplementary Figs. 9–11. Error bars represent the mean ± SD of four (**a** and **b**) or six (**c**) independent experiments. The single and double asterisk represents $p < 0.05$ and $p < 0.01$, respectively. The $p$-values of the two-sided Welch's $t$ test are provided in Source Data. TATA-box and spacer sequence are indicated in red and gray boxes, respectively. Fluorescence measurement for each iSynP is shown in Supplementary Fig. 8. DAPG, 2,4-diacetylphloroglucinol; Dox, doxycycline; EGFP, enhanced green fluorescent protein; HSL, *N*-(ketocaproyl)-D,L-homoserine lactone. ns, not significant.

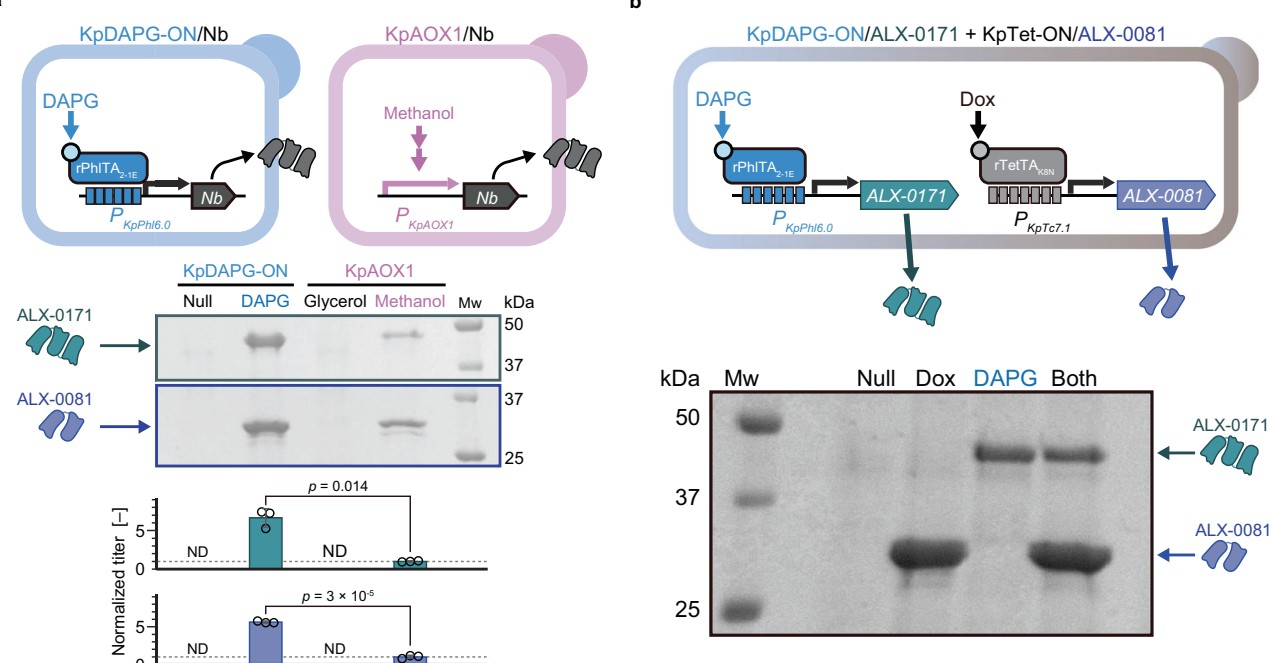

**Fig. 5 | Inducible production of nanobodies using conventional promoter or iSynPs. a** Inducible production of ALX-0081 ($n = 3$). **b** Selective production of different nanobodies using a single yeast strain ($n = 3$). DAPG and Dox were used at concentrations of 10 μM and 30 μg/L, respectively. Error bars represent the mean ± SD of three independent experiments. The $p$-values of the two-sided Welch's $t$ test are indicated. DAPG, 2,4-diacetylphloroglucinol; Dox, doxycycline; Mw, molecular weight marker; Nb, nanobody; ND, not determined. Detailed information on the strains is indicated in Supplementary Figs. 14 and 15.

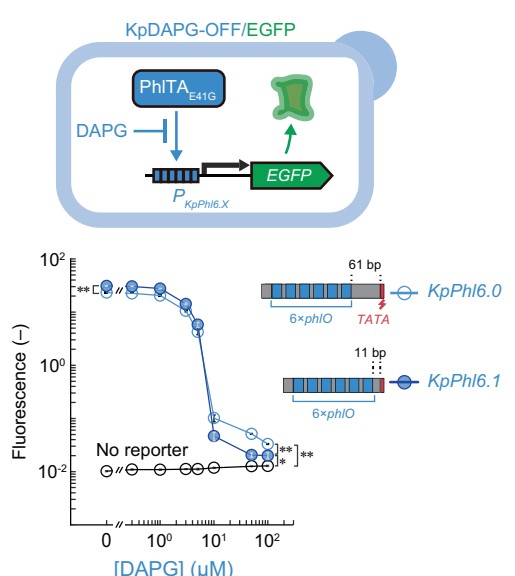

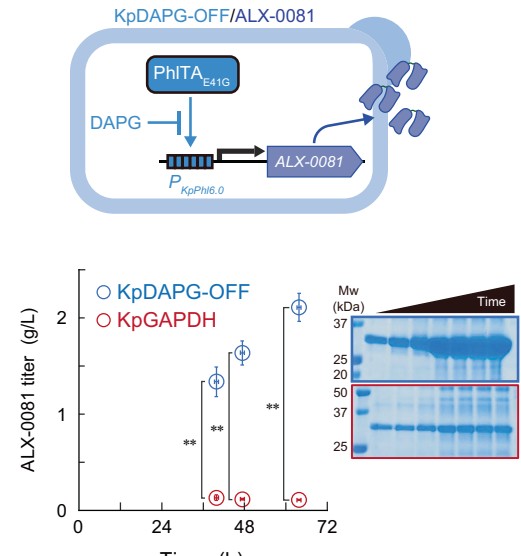

**Fig. 6 | Inducer-free expression platform for single biologics production via large-scale fed-batch fermentation. a** Schematic of the DAPG-deactivatable inducible system (KpDAPG-OFF) and its control of EGFP expression. **b** Five-liter fed-batch fermentation of ALX-0081 using the KpDAPG-OFF system with a single expression cassette. To repress ALX-0081 expression, DAPG was added to a 100 mL preculture at a final concentration of 10 μM. A single-copy expression cassette with the *KpGAPDH* promoter driving ALX-0081 expression was used as a control.

Glucose feeding rates for each fermentation are shown in Supplementary Fig. 17. Uncropped gel images are additionally provided in Source Data. Detailed information on the strains is indicated in Supplementary Fig. 18. Error bars represent the mean ± SD of four (**a**) or three (**b**) independent experiments. The single and double asterisk represents $p < 0.05$ and $p < 0.01$, respectively. The *p*-values of the two-sided Welch's *t*-test are provided in Source Data. DAPG, 2,4-diacetylphloroglucinol; EGFP, enhanced green fluorescent protein; Mw, molecular weight marker.

## Inducer-free expression system for large-scale fermentation

"Inducer-OFF" switches, in which target gene expression is deactivated or repressed in the presence of an inducer and activated upon inducer removal, are preferred to avoid the cost of inducers, especially in large-scale fermentation[32]. For these systems, we used the previously constructed DAPG-deactivatable transcription activator, PhlTA[13]. To re-optimize this PhlTA for the highly inducible promoter with six *phlO* repeats, a single round of directed evolution of PhlTA was performed using our recently reported evolutionary platform[13] (Supplementary Fig. 16a). Briefly, PCR-randomized *phlTA* was cloned into an expression vector in the *S. cerevisiae* strain harboring the iSynP (six copies of *phlO* fused with *ScGAL1* core promoter, [$P_{ScphlO6}$]) followed by the trifunctional fusion selector gene (TB$_{D25A}$G), encoding herpes simplex virus thymidine kinase, D25A mutant of Zeocin resistance marker (BLE$_{D25A}$), and monomeric umikinoko green 1 (mUkG1). The resulting yeast library was first subjected to ON selection in the absence of DAPG; yeast variants with strongly activated $P_{ScphlO6}$ by the mutant PhlTA could survive under Zeocin selection as BLE$_{D25A}$ inactivates Zeocin (ON selection). Subsequently, in the presence of 5 or 0.5 μM DAPG, yeast variants expressing TB$_{D25A}$G via the incomplete deactivation of mutant PhlTA by DAPG caused cell death in the presence of 5-Fluoro-2-deoxyuridine because hsvTK convert it to the toxic compound (OFF selection). From the resulting two selected yeast libraries, a total of 93 clones were ON/OFF screened by measuring the presence or absence of 5 μM DAPG. DNA sequence analysis of the screened mutants identified one beneficial mutation in PhlTA (E41G) (Supplementary Fig. 16b). Introducing this mutant PhlTA (PhlTA$_{E41G}$) into *K. phaffii* harboring iSynPs with the six *phlO* resulted in a 702- and 1572-fold reduction of GFP expression upon DAPG addition with 61- and 11-bp spacers upstream of the TATA-box, respectively (Fig. 6a). The induction system was subjected to bioreactor fermentation for the production of ALX-0081 (Fig. 6b). Using the iSynP with a 61-bp spacer upstream of the TATA-box ($P_{KpPhl6.0}$),

approximately 2 g/L ALX-0081 was secreted into the supernatant of the yeast culture, which was >10-fold higher compared to the conventional *KpGAPDH* promoter system.

## Production of a difficult-to-express virus antigen for animal immunization

We evaluated whether our expression platform can be applied not only to produce well-engineered nanobodies like ALX-0081 but also to produce "difficult-to-express" proteins, such as virus antigen proteins. Accordingly, we attempted to produce an RBD from a SARS-CoV-2 omicron variant (RBD$^{om}$) harboring 15 mutations known to reduce stability and expression[19,33], using the DAPG-inducible (KpDAPG-ON) system. The gene encoding RBD$^{om}$ was cloned downstream of the DAPG-iSynP insulated with a ca. 1-kb *KpARG4* sequence and integrated into the *K. phaffii* strain expressing rPhlTA$_{2-1E}$. The resulting yeast strain produced an RBD$^{om}$ variant when incubated at 18 °C rather than 30 °C (Fig. 7a), consistent with the reported instability of RBD$^{om}$ [19,33], yielding 2.5-mg purified protein when cultured in 225 mL in test tubes (Supplementary Fig. 19). Mass spectrometric analysis of the purified and deglycosylated RBD$^{om}$ stored at 4 °C revealed that full-length antigen was produced despite N- and C-terminal heterogeneity resulting from incomplete signal peptide cleavage[34] and nonspecific proteolysis, respectively (Fig. 7b–d and Supplementary Fig. 19). The purified RBD$^{om}$ was used for chicken immunization, followed by phage-based biopanning cycles using a commercially obtained RBD$^{om}$ (Fig. 8a). During each round of biopanning, phage variants that bind to a wild-type RBD produced with the recombinant yeast were removed to obtain antibodies that can distinguish RBD$^{om}$ from wild-type RBD. From the enriched library of single-chain Fv variants, we selected one clone (#Omi-7) and recombined it into an immunoglobulin Y (IgY) antibody. The specificity of this IgY antibody toward RBD$^{om}$ against wild-type RBD was evaluated using the antibody to detect RBD$^{om}$ by enzyme-linked immunosorbent assay (ELISA) (Fig. 8b) and western blot analysis

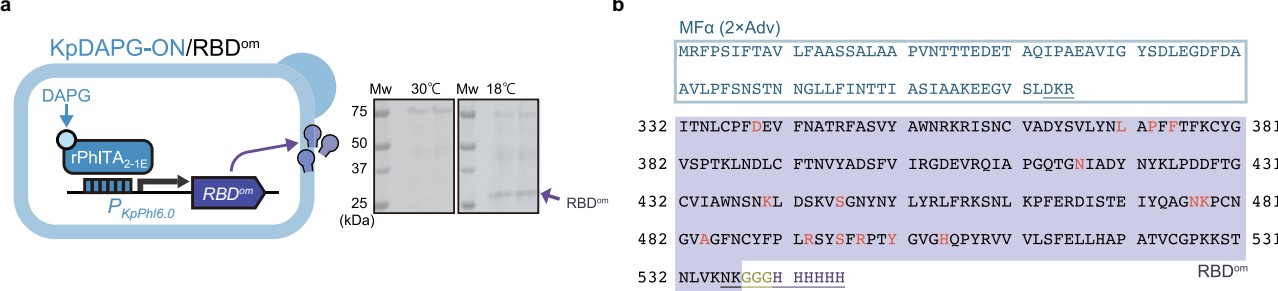

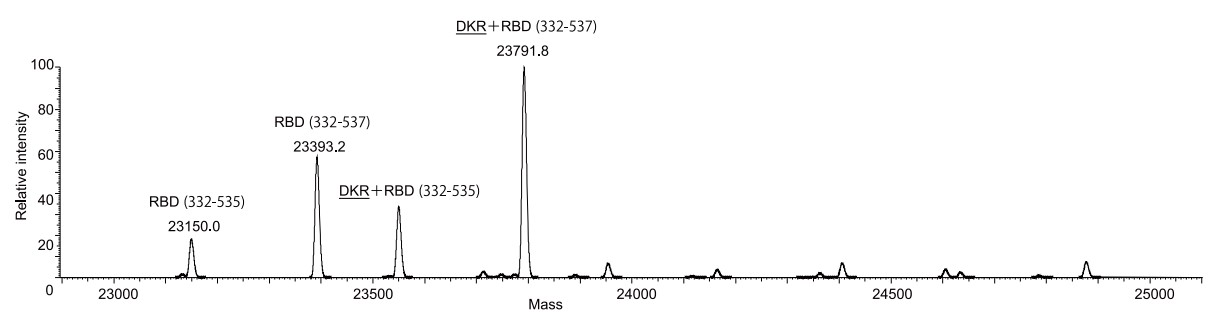

**Fig. 7 | Yeast-based production of RBD protein from a SARS-CoV-2 omicron variant using the DAPG-inducible expression platform. a** Production of secreted RBD protein from the SARS-CoV-2 omicron variant (RBDᵒᵐ) using the KpDAPG-ON system. Following incubation at 30 °C and 18 °C in 2 mL of buffered-minimal glucose yeast (BMDY) media containing 50 g/L hipolypeptone and 10 µM DAPG, the yeast supernatant was analyzed by SDS-PAGE (*n* = 2). Detailed information on the strain is indicated in Supplementary Fig. 19a. **b** Amino acid sequence of RBDᵒᵐ protein. The signal sequence (MFα, 2 × Adv) is highlighted in indigo. Omicron mutations are highlighted in red. Terminal heterogeneous sequences are under-lined. **c**, **d**. N- and C-terminal heterogeneity analysis of RBDᵒᵐ protein purified from the supernatant of engineered yeast cells (**a**). **c** Major N- and C-terminal amino acid sequences were detected by peptide mapping with chymotrypsin digestion. A confidence score of 100% indicates an ideal fit between the predicted spectra and the experimental spectra. **d** Typical deconvolution mass spectrum obtained by intact mass analysis. Mw, molecular weight marker.

(Fig. 8c). The results indicated that the yeast-based platform could produce sufficient amounts of virus antigens for chicken immunization even with a small-scale fermentation in test tubes.

## Discussion

Regardless of its source, any sequence can potentially act as a weak activator when placed upstream of a TATA box in an iSynP, leading to unintended basal expression (leakiness). This cryptic activation is difficult to predict, but we have shown that leakiness from iSynPs could be largely mitigated by insulation (Fig. 2), increasing operator repeats (Fig. 4 and Supplementary Fig. 8), and mutating causal operators (Fig. 4c and Supplementary Fig. 12b). Although mutating BOs may inhibit the binding in some cases, the DNA binding of bacterial transcription factors to mutant operators may be improved through domain swapping[35] or fast-directed evolution strategies[13,36,37]. By effectively mitigating leakiness, we can achieve strong iSynP construction through the direct fusion of bacterial operator repeats to the TATA-box sequence. Considering all our findings, we established a generalizable design for creating potent iSynPs in various eukaryotic

yeasts (Fig. 1). Although large DNA constructs could be more stably assembled in yeasts[38], adding more than 1-kb insulating sequence upstream of each iSynP is a potential drawback for construction of multigene pathways or complex, higher-order genetic circuits. Comprehensive studies that could identify the "sequence-leakiness relationship" could further simplify the iSynP design.

Because of the leak-prone nature of yeast iSynPs, their minimum sequence requirements remain unclear. Subsequently, establishing well-insulated iSynPs with negligible leakiness provided an in-depth insight into how to construct minimal, strong iSynPs by testing a series of iSynPs with different architectures (Fig. 3). Surprisingly, we found that only a 94-bp sequence could work as a promoter to induce downstream reporter expression >10³-fold, which outperforms recently reported strong yeast iSynPs[1,4,13,39], although they might not be compared directly. Moreover, the induced promoter activity was stronger compared to the conventionally strongest natural constitutive promoter in yeast (Fig. 3c). This promoter consists of three essential elements: a 30-bp bacterial operator fused to a 9-bp TATA-box sequence with an 10-bp spacer and a downstream 45-bp sequence,

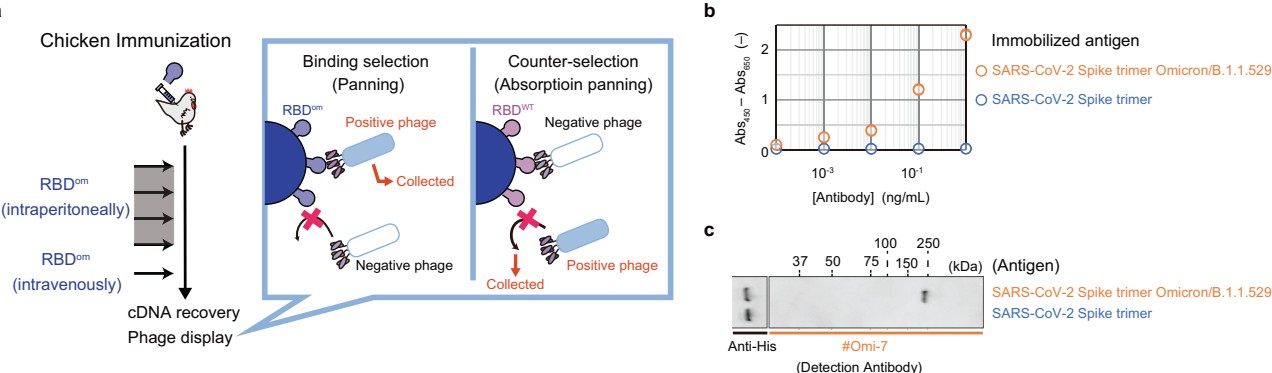

**Fig. 8 | Generation of specific antibody toward RBD protein from a SARS-CoV-2 Omicron variant by immunizing chickens with a yeast-produced antigen.** **a** Diagram of chicken immunization with RBD$^{om}$ protein prepared with yeast. IgY antibody was isolated from chicken immunization, followed by phage-based bio-panning. **b**, **c** The recombinant IgY antibody (#Omi-7) was tested for specific binding to the spike trimer protein from the wild-type and Omicron variant of SARS-CoV-2 using ELISA (**b**) and western blotting (**c**). **b** ELISA: Spike trimer protein from the wild-type or omicron variant of SARS-CoV-2 was immobilized onto plates, followed by the successive addition of IgY #Omi-7 antibody and HRP-labeled anti-chicken-IgG antibody to evaluate its specific binding to the immobilized proteins. Individual ELISA signals of two independent experiments are indicated with dots. **c** Western blot analysis of spike trimer protein from the wild-type or omicron variant of SARS-CoV-2 ($n = 1$). These proteins were detected using IgY #Omi-7 and HRP-labeled anti-chicken IgG as primary and secondary antibodies, respectively, or using HRP-conjugated anti-His-tag antibody as a positive control. Abs$_{450}$, absorbance at 450 nm; Abs$_{650}$, absorbance at 650 nm.

including a transcription start site, resulting in one of the shortest yeast iSynPs ever reported.

The constructed iSynPs were used to produce two pharmaceutical proteins (Figs. 5, 6). The protein production platform using our iSynPs is characterized by the flexibility of the incubation conditions, such as carbon source, and the capability for point-of-care production of different proteins with only single-engineered yeast. Specifically, we prepared a vaccine candidate protein for SARS-CoV-2 in sufficient yield for chicken immunization to produce antibodies, even without optimizing the fermentation and purification steps (Figs. 7, 8). Although the incomplete signal peptide cleavage and the nonspecific proteolysis during the storage period remain to be addressed for future studies, the former may be alleviated by inserting an appropriate linker[40]. Despite the difficulties in producing RBD from a SARS-CoV-2 omicron variant with conventional expression systems[19], our results demonstrate rapid RBD production without using a high-cell density fermentation using a jar-fermenter, indicating the robustness of the yeast-based expression platform for vaccine production against new pandemic viruses.

Regarding the economic viewpoint, the cost for DAPG induction (10 μM) is approximately twofold higher than conventional inducer (methanol, 1 [v/v]%). Applying the design principle to construct inducible systems described in this study, other bacterial induction systems with cheaper induction molecules, such as choline chloride[41], could be used to save the induction costs. Alternatively, the DAPG-OFF system, which enables the induction by the removal of DAPG, could be used to minimize the induction costs. Taken together, our findings demonstrate the usefulness of our gene expression platforms in yeast. This will expand our capability to engineer yeast functions as intended and boost various synthetic biology projects.

## Methods

### Ethical statement
All animal experiments were performed in accordance with the Guidelines for the Care and Use of Laboratory Animals of Pharma Foods International Co. Ltd. Experimental protocols and studies were reviewed and approved by the Animal Experimentation Management Committee of Pharma Foods International Co., Ltd.).

### Media and buffers
YPD medium (10 g/L yeast extract [Nacalai Tesque, Kyoto, Japan], 20 g/L Bacto Peptone [BD Biosciences, San Jose, CA, USA], and 20 g/L glucose), SD + Met medium (6.7 g/L BD Difco yeast nitrogen base (YNB) without amino acids [BD Biosciences], 20 μg/mL L-methionine, and 20 g/L glucose), and SCSM-His-Leu-Ura medium (6.7 g/L YNB, 1.656 mg/L SCSM-His-Leu-Ura mixture [Cat. No. DSCS251, ForMedium, Norfolk, UK]), and 20 g/L glucose) were used to culture yeast cells. For biological fermentation, buffered-minimal glycerol yeast extract (BMGY), buffered-minimal methanol yeast extract (BMMY), and BMDY media were prepared by adding 20 g/L of glucose, methanol, and glycerol, respectively, to a basal medium composed of 10 g/L yeast extract, 20 g/L hipolypeptone [Nihon Pharmaceutical, Tokyo, Japan], 13.4 g/L YNB, 0.4 mg/L biotin [Nacalai Tesque], and 100 mM potassium phosphate buffer [pH 6.0]. Next, Luria-Bertani (LB) medium (25 g/L LB Broth, Miller [Nacalai Tesque]) containing 100 μg/mL ampicillin was used to culture *Escherichia coli*. Antibiotics (500 μg/mL G418 [Fujifilm Wako Pure Chemical, Osaka, Japan], 100 μg/mL Zeocin™ [InvivoGen, San Diego, CA, USA], 50 μg/mL nourseothricin [Werner BioAgents, Jena, Germany], 500 μg/mL blasticidin [Fujifilm Wako Pure Chemical] and 300 μg/mL hygromycin [Fujifilm Wako Pure Chemical]) were used to select stably transformed yeast strains. Appropriate amounts of HSL (Cat. No. FK29472, Carbosynth, San Diego, CA, USA) and Dox (Cat. No. 631311, Clontech Laboratories, Mountain View, CA, USA) were dissolved in dimethyl sulfoxide (DMSO) (Cat. No. 13445-74, Nacalai Tesque) and water, respectively, to prepare stock solutions. DAPG (Cat. No. Sc-206518, Santa Cruz Biotechnology, Dallas, TX, USA) was dissolved in ethanol or DMSO to prepare a stock solution. Because DAPG has been previously found to be light-sensitive[42], the stock DAPG solution was added to the media just before use, and all resultant media were stored in the dark. Pichia trace metal 1 (PTM1) solution (containing 6.0 g/L CuSO$_4$·5H$_2$O, 0.08 g/L NaI, 3.0 g/L MnSO$_4$·H$_2$O, 0.2 g/L Na$_2$MoO$_4$·2H$_2$O, 0.02 g/L H$_3$BO$_3$, 0.5 g/L CoCl$_2$, 20 g/L ZnCl, 65 g/L FeSO$_4$·7H$_2$O, 0.2 g/L biotin, and 5.0 mL H$_2$SO$_4$) was used for 5-L fed-batch fermentation. PBS buffer was prepared by diluting a 10 × PBS buffer stock (Cat. No. 7575-31, Nacalai Tesque). PBS-T buffer (containing 137 mM NaCl, 8.9 mM Na$_2$HPO$_4$·12H$_2$O, 2.7 mM KCl, 1.5 mM KH$_2$PO$_4$, and 0.05% Tween-20) and TBS-T buffer (50 mM Tris-HCl [pH 7.6] and 0.05% Tween-20) were used for ELISA and western blotting, respectively.

### Strains, primers, and plasmids
The *E. coli* strain DH5α was used for plasmid subcloning. Here, the *K. phaffii* strain CBS 7435 (NRRL-Y11430, ATCC [Manassas, VA, USA]) and

the two *S. cerevisie* strain BY4741-*Phl₂-₁E*-*Lux₂-₄F*[13] and BY4741-*hENT1*[13] were used as a parental yeast strain. The plasmids were individually linearized using appropriate restriction endonucleases and transformed into yeast cells by either the conventional chemical method[43] or electroporation[44]. Successful integration of the plasmid in the desired configuration was verified through colony-direct PCR. Detailed information on all plasmids, yeast strains (*K. phaffii* and *S. cerevisiae*), primers, DNA parts, iSynPs, and insulators employed in this study are presented in Supplementary Data 1–7. All yeast strains except for those harboring centromeric plasmids were stored as glycerol stock and streaked onto the YPD plate before experiments. All experiments were performed using the resulting colonies, with at least three biological replicates unless otherwise noted. The following plasmids were used for plasmid construction: pGK415-*phlTA*[13], pGK415-*phlTA₁-₂E*[13], pGK415-*rphlTA₂-₁E*[13], pGK401red-*rtetTA*$_{K8N, L131L}$[13], pRS406red-$p_{phlO1}$ -*TBG*[13], pGK416m-$p_{phlO6}$ -*ymUkG1*[13], pGK416m-$p_{tetO7}$ -*ymUkG1*[13], pGK416m-$p_{luxO5}$ -*ymUkG1*[13], pGK416-*ymUkG1*[45], pUC19-MCS-Zeo[30], pPGP_EGFP[46], pNTI647[47], pYN169[48], and pYN186[48]. pNTI647 dCas9-Mxi1 TetR KanMX was a gift from Dr. Nicholas Ingolia (Addgene plasmid # 139474)[47].

### Fluorescence analysis with flow cytometry
Fluorescence analysis of yeast was performed using a previously described method[49] with slight modifications. Briefly, *K. phaffii* strains were grown in 500 μL/well YPD media on 96-well plates (Costar®, #3960 [Corning, Corning, NY, USA]) with different concentrations of inducers (10 μM for DAPG, 30 μg/mL for Dox, and 3 μM for HSL, unless otherwise noted) for 24 h (30 °C, 1000 rpm). For the KpDAPG-OFF system, 30 μM DAPG was added to the preculture media. The resulting yeast cells in the stationary phase were harvested and analyzed using a CytoFLEX (Beckman Colter, Brea, CA, USA) flow cytometer with a 488 nm laser and a bandpass filter for GFP (525/40 nm). Median fluorescence and forward scatter intensity measurements (i.e., FITC-A and FSC-A) were taken. Cell-size normalized fluorescence was then calculated by dividing the FITC-A signal by the FSC-A signal. Fold induction was calculated by dividing the normalized fluorescence of induced cells by that of uninduced cells. For *S. cerevisiae*, SCSM-His-Leu-Ura (Supplementary Fig. 7) or SD + Met (Supplementary Fig. 16) media were used, and mean fluorescence and forward scatter intensity were taken to calculate the cell-size normalized fluorescence.

### SDS-PAGE analysis
An equivalent of 50 μL of the supernatant obtained by centrifugation of each yeast cell culture was mixed with 10 μL of 6 × sample buffer containing a reducing reagent (Cat. No. 09499-14, Nacalai Tesque). This was followed by incubation at > 95 °C for more than 10 min. Next, 15 μL of the resultant lysate was separated on a 15% polyacrylamide gel using SDS-PAGE (e-PAGEL [Cat. No. E-R15L, ATTO, Tokyo, Japan]) or 5–20% gradient polyacrylamide gel (m-PAGEL [Cat. No. M-520L, ATTO]) using SDS-PAGE and stained with CBB Stain One (Nacalai Tesque).

### Small-scale nanobody production
For the production of ALX-0081 using iSynP or $P_{KpAOX1}$, cells were grown overnight at 30 °C and 170 rpm in BMDY (for iSynP) or BMGY (for $P_{KpAOX1}$) media and collected by centrifugation. After washing, 5% of the cell pellet was resuspended in 2 mL of BMDY medium supplemented with DAPG (10 μM) for DAPG induction and BMMY medium for methanol induction, and incubated for 60 h. For the selective production of two nanobodies with single yeast, 10% of the cell pellet prepared with the overnight yeast cell culture (30 °C, 170 rpm) in BMDY medium was resuspended in 2 mL of BMDY media supplemented with DAPG (10 μM) and Dox (30 μg/mL) in different combinations and incubated for 49 h to induce nanobody secretion. After incubation, the cell pellet was harvested via centrifugation, and 50 μL

of the supernatants were then subjected to SDS-PAGE analysis to evaluate nanobody production per volume.

### Directed evolution of PhlTA in *S. cerevisiae*
The directed evolution of PhlTA was performed as described previously[13]. Briefly, the wild-type *phlTA* gene along with short untranslated sequences (i.e., 104-bp upstream and 85-bp downstream sequence of *phlTA*) was amplified via error-prone PCR using Takara *Taq*$^{TM}$ DNA polymerase (Takara Bio, Shiga, Japan) and MnCl₂ (50 μM, Nacalai Tesque) using the MT415 and MT416 primers (see Supplementary Data 4). The resulting fragment and the digested expression vector were used to transform yeast strain ScMT487, wherein these two DNA fragments were assembled via GAP-repair cloning. The resulting transformants (equivalent to $4 \times 10^4$ colonies) were recovered from the SD + Met media and subjected to three successive incubations: first in the presence of 4.8 mg/mL Zeocin, then in the presence of 5 or 0.5 μM DAPG, and finally with both 5 or 0.5 μM DAPG and 4.8 mg/mL Zeocin. Between each incubation step, the yeast cell populations were mixed with glycerol and stored at − 80 °C until subsequent analysis. Selected cells were then propagated on agar plates, and individual clones were incubated both in the presence and absence of DAPG, followed by flow cytometry analysis.

### Fed-batch fermentation for secreted protein production
A 5-L fed-batch fermentation was performed following a previously described protocol[48], with some modifications. In particular, BMDY media was used throughout the process, including seed culture preparation. Briefly, a preculture was incubated with 100 mL BMDY media in a 1-L Erlenmeyer flask at 30 °C under shaking at 180 rpm and was then transferred to 1.5-L BMDY media in a 5-L fermenter (BMS-P; ABLE & Biott Co., Ltd., Tokyo, Japan) (initial $OD_{660}$~ 1.0). This was followed by fermentation under the following conditions: dissolved oxygen (DO), temperature, and pH set at 30%, 30 °C, and 6.0, respectively. Automatically controlled stirring at 200−800 rpm using a Proportional Integral-Differential Controller (DPC-4; ABLE & Biott Co., Ltd.) and compressed 98% O₂ gas (ITO-08-1(2); IBS Co. Ltd., Osaka, Japan) were used to maintain a constant DO level. Glucose solution (450 g/L) supplemented with 12.0 mL/L PTM1 solution was fed to the medium at different feeding rates ranging from 0.2 to 1.2 g/min (Supplementary Fig. 17). $OD_{660}$ was monitored at each time point, and the protein levels in the supernatant were determined by SDS-PAGE. Bovine serum albumin solutions were used as calibration standards at various concentrations (120, 60, 30, 15, and 7.5 mg/L).

### Preparation of RBD$^{om}$ via yeast fermentation
The recombinant yeast colony was inoculated into 25 mL BMDY media containing 50 g/L hipolypeptone in nine large test tubes (Cat. No. 245920, EYELA, Tokyo, Japan) and incubated overnight (30 °C, 170 rpm). The supernatant was then replaced with the same volume of fresh medium supplemented with 10 μM DAPG and incubated for another 60 h (18 °C, 170 rpm). Subsequently, 225 mL supernatant was collected by centrifugation (2380 × *g*, 5 min), and the His-tag protein was column-purified using two sets of Capturem His-tag Purification Maxiprep Kits (Takara Bio), followed by desalination in PBS buffer (Cat. No. 7575-31, Nacalai Tesque) using Prepacked Disposable PD-10 columns (Code No. 17085101, Cytiva, Marlborough, MA, USA). The recovered protein concentration was then determined using a Qubit$^{TM}$ Protein Assay Kit (Code No. Q33212, Thermo Fisher Scientific, Waltham, MA, USA) and a Qubit® 2.0 Fluorometer (Thermo Fisher Scientific) before being stored at 4 °C until use. Before peptide mapping and intact mass analysis, the stored sample was subjected to gel filtration chromatography using Superdex 200 increase (Cytiva) and concentrated 10-fold using Amicon Ultra-15 Ultracel-10K (Cat. No. UFC901096, Merk Millipore, MA, USA). This process was performed twice and then used for analysis.

## Peptide mapping and intact mass analysis of RBD^om by liquid chromatography/mass spectrometry (LC/MS)

Peptide mapping and intact mass analysis via LC/MS were performed using a Vanquish UHPLC System (Thermo Fisher Scientific) and an Orbitrap Fusion Lumos Tribrid mass spectrometer (Thermo Fisher Scientific). For peptide mapping, the sample was prepared as previously reported[50,51], with slight modifications. Briefly, after reduction and carboxymethylation, the RBD^om protein (10 μg) was deglycosylated overnight at 37 °C using 10 units of PNGaseF (Roche, Mannheim, Germany). Subsequently, the carboxymethylated protein was digested using 2 μg of chymotrypsin (sequencing grade, 1 mg/mL; Promega, Madison, WI, USA) at 37 °C overnight. This was followed by desalting using an Oasis HLB μElution plate (Waters, Manchester, UK). A sample of the digested peptide solution was analyzed via LC/MS using data-dependent higher-energy collisional dissociation. The resulting peptide fragments were then identified using BioPharma Finder 3.1 (Thermo Fisher Scientific). Precursor mass tolerance was set to ± 5 ppm. Carboxymethylation (+ 58.005 Da) was set as a static modification of cysteine residues. Oxidation (+ 15.995 Da) of methionine, tryptophan, and tyrosine, and deamidation (+ 0.984 Da) of asparagine and glutamine were set as variable modifications. For intact mass analysis, RBD^om protein (10 μg) was deglycosylated for 3 days at 37 °C using 1 μL of PNGaseF (Waters, Milford, MA, USA), and analyzed via LC/MS on a polyphenyl column (Waters, BioResolve RP mAb, 2.7 μm, 2.1 × 150 mm) with a column temperature of 30 °C. Mobile phases A and B were prepared by adding 0.1% formic acid (Kanto Chemical, Tokyo, Japan) to distilled water and acetonitrile, respectively. Proteins were then eluted using a linear gradient of mobile phase B from 5% to 90% over 15 min at a flow rate of 200 μL/min. The parameters for mass spectrometry were as follows: electrospray voltage, 3.5 kV in positive ion mode; ion transfer temperature, 300 °C; full mass scan range, $m/z$ 500–4500; full mass scan Orbitrap resolution, 15,000; and in-source collision voltage, 30 V. Finally, proteins were identified using BioPharma Finder 3.1 (Thermo Fisher Scientific). The search parameters were as follows: a merge tolerance of 10 ppm, a deconvolution mass tolerance of 10 ppm, and a peak detection quality measure of 95%. Mass spectrometric data have been deposited at ProteomeXchange/jPOST[52].

## Chicken immunization of RBD protein using a SARS-CoV-2 omicron variant

Chicken immunization was performed as described previously with some modifications[53]. Briefly, three female chickens of 34-day-old (Boris Brown, Mie Hiyoko Co. Ltd., Mie, Japan) were used for immunization. The antigen was mixed with Freund's Complete Adjuvant (Cat. No. 014-09541, Fujifilm Wako Pure Chemical) or Freund's Incomplete Adjuvant (Cat. No. 011-09551, Fujifilm Wako Pure Chemical) for first and subsequent immunizations, respectively, together with D-PBS (Cat. No. 045-29795, Fujifilm Wako Pure Chemical). A total of five immunizations were performed, in which 1 mL of mixed antigen per dose was administered intraperitoneally (from the first to the fourth immunizations) and intravenously (for the fifth immunization). Three days after the final immunization, the immunized chickens were euthanized, and the spleens were collected. Total RNA was extracted from the spleens, and cDNA was obtained via reverse transcription using the PolyATtract® mRNA Isolation Systems (Cat. No. Z5310, Promega) and the PrimeScript II 1st Strand cDNA Synthesis Kit (Cat. No. 6210 A, Takara Bio).

## Phage-based biopanning

Using the above cDNA mixtures, a scFv phage library was constructed using a previously described method[53] and used for a panning assay[54]. Briefly, the scFv DNA fragments were cloned into a phagemid vector, and the resulting library plasmid was electroporated into *E. coli* XL1-Blue Electroporation Competent Cells (Cat. No. 200228, Stratagene,

La Jolla, CA, USA). The recovered *E. coli* transformants were then incubated with VCSM13 helper phage (Cat. No. 200251, Stratagene) at 37 °C with vigorous shaking, and the medium was replaced with one containing isopropyl β-D-thiogalactopyranoside, and the culture was incubated at 30 °C overnight. After incubation, the supernatant was filtrated to obtain a phage display library. For their selection, 50 μL (final conc. 5 μg/mL) of RBD^om (Cat. No. 40592-V08H121, Sino Biological Inc., Beijing, China) was coated on a Maxisorp Nunc-Immuno™ plate (Cat. No. 442404, Thermo Fisher Scientific). At each round, absorption panning to deplete scFv variants that were bound to the domain was performed by using wild-type RBD produced with the recombinant *K. phaffii* strain PpYI1924 (Supplementary Data 2). After the 11th repetition of the panning procedure, scFv variants that showed strong binding to RBD^om but not to wild-type RBD were isolated and sequenced to identify complementarity-determining region (CDR) sequences. Full-length scFv sequences were sequenced via sanger-sequencing using primer 5´-GGAAA-CAGCTATGACCATG-3´ at Eurofins Genomics Inc. (Tokyo, Japan), and each CDR sequence was identified according to the Kabat numbering scheme. Based on these sequencing data, the DNA sequences coding the heavy and light chains were cloned into a transient expression vector for transformation into Expi293F cells (Cat. No. A14528, Thermo Fisher Scientific). The resulting secreted IgY antibody was then purified using Ni-Sepharose Excel (Cat. No. 17-3712-01, Thermo Fisher Scientific) and desalted on a PD-10 column (Cytiva) using PBS buffer.

## ELISA

ELISA was performed as described previously[54]. Briefly, plate wells were coated with 50 μL (1 μg/mL) of SARS-CoV-2 Spike trimer (Cat. No. SPN-C52H9, Acro Biosystems, Tokyo, Japan) or SARS-CoV-2 Spike trimer Omicron/B.1.1.529 (Cat. No. SPN-C52Hz, Acro Biosystems), followed by blocking for 1 h at 37 °C with 25% BlockAce in PBS. Next, the different concentrations of monoclonal IgY antibody were added to each well of the plate, followed by incubation at 37 °C for 1 h. The bound antibodies were then washed with PBS-T buffer and incubated with 500 ng/mL HRP-anti-chicken IgG (Cat. No.5220-0373, SeraCare Life Sciences, Milford, MA, USA) for 1 h at 37 °C. After washing with PBS-T, SureBlue™ (Cat. No. 5120-0074, SeraCare Life Sciences) was added and incubated for 30 min. Finally, the TMB stop solution (Cat. No. 5150-0019, SeraCare Life Sciences) was added to stop the reaction. Absorbance was measured at 450 and 650 nm using a microplate reader (Cytation 5 imaging reader from BioTek, Winooski, VT, USA).

## Western blotting

An equivalent of 10 ng SARS-CoV-2 Spike trimer (Cat. No. SPN-C52H9, Acro Biosystems) and SARS-CoV-2 Spike trimer Omicron/B.1.1.529 (Cat. No. SPN-C52Hz, Acro Biosystems) were analyzed via SDS-PAGE using a NuPAGE 4–12% Bis-Tris gel (Cat. No. NP0329BOX, Thermo Fisher Scientific) and MOPS buffer (Cat. No. NP0001, Thermo Fisher Scientific). The proteins were then transferred to iBlot2 PVDF Mini Stacks (Cat. No. IB240002, Thermo Fisher Scientific) using an iBlot2 Gel Transfer Device (Cat. No. IB21001, Thermo Fisher Scientific). After blocking with Blocking One solution (Cat. No. 03953-95, Nacalai Tesque) for 30 min at room temperature, the membrane was incubated with #Omi-7 IgY (100 ng/mL) diluted in the same blocking buffer at 4 °C overnight. The membranes were then washed with TBS-T buffer (5 min, three times) before being incubated with a secondary antibody (HRP-anti-chicken-IgG, 50 ng/mL in blocking solution) at room temperature for 1 h, followed by additional washing in TBS-T. Finally, the membrane was developed using ECL™ Prime Western Blotting Detection Regents (Cat. No. RPN2232, Cytiva), followed by analyzing the immunoreactive bands using a LuminoGraph (WSE-6100, ATTO). The His-tag proteins were detected using an HRP-conjugated His-tag

antibody (Cat. No. HRP-66005, Proteintech, Rosemont, IL, USA) instead of primary and secondary antibodies.

## Statistics and reproducibility

Statistical difference ($p$-value) was determined with a two-tailed Welch's or paired $t$ test. All yeast strains except for $S. cerevisiae$ strains harboring centromeric plasmids were stored as glycerol stock and streaked onto the YPD plate before experiments. Experiments were performed with the resulting colonies in more than triplicate unless otherwise noted. No statistical method was used to predetermine sample size, and no data were excluded from the analyses otherwise noted.

## Reporting summary

Further information on research design is available in the Nature Portfolio Reporting Summary linked to this article.

## Data availability

The source data underlying Figs. 2–8 and Supplementary Figs. 1, 4, 5, 7, 8, 12, 13, 16, and 17 are provided as a source data file. The following plasmids are available through Addgene: pKN108 (#228284), pYI277 (#228285), pMT478 (#228286), pMT856 (#228287), pMT899 (#228288), pMT949 (#228289), pMT950 (#228290), pMT951 (#228291), pMT952 (#228292), pMT957 (#228293), pMT958 (#228294), pMT959 (#228295), pMT960 (#228296), pMT961 (#228297), pMT1014 (#228298), pMT1026 (#228299), pMT1035 (#228300), pMT1219 (#228301), pMT1328 (#228302), pMT1341 (#228303), pMT1342 (#228304), pMT1343 (#228305), pMT1344 (#228306), pMT1345 (#228307), pMT1346 (#228308), pMT1347 (#228309), pMT1348 (#228310), pMT1349 (#228311), pMT1350 (#228312), pMT1351 (#228313), pMT1375 (#228314), pMT1419 (#228315), pMT1420 (#228316), pMT1434 (#228317), pMT1449 (#228318), pMT1461 (#228319), pMT1473 (#228320), pMT1474 (#228321), pMT1475 (#228322), pMT1481 (#228323), pMT1482 (#228324), pMT1483 (#228325), pMT1487 (#228326), pMT1821 (#228327), pMT1856 (#228328), pMT1857 (#228329), pMT1858 (#228330), pMT1876 (#228331), pMT1877 (#228332), pMT1883 (#228333), pMT1980 (#228334), pMT1981 (#228335), pMT2013 (#228336), pMT2076 (#228337), pMT2077 (#228338), pMT2080 (#228339), pMT2081 (#228340), pMT2086 (#228341), pMT2092 (#228342), pMT2093 (#228343), pMT2094 (#228344), pMT2095 (#228345), pMT2096 (#228346), pYI2165 (#228347), pYI2206 (#228348), pYI2216 (#228349), pYI2292 (#228350), pYI2293 (#228351), pYI2294 (#228352). Mass spectrometric data have been deposited at ProteomeXchange/jPOST and are publicly available with the identifier PXD057342 for ProteomeXchange and JPST003442 for jPOST. Source data are provided in this paper.

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

## Acknowledgements

We thank Aki Ichimichi, Yuko Kawasaki, and Yoshimi Tsuruma for their technical assistance. This study was supported by the Project Focused on Developing Key Technology for Discovering and Manufacturing Drugs for Next-Generation Treatment and Diagnosis from the Japan Agency for Medical Research and Development (AMED), Japan (Grant Numbers: JP21ae0121002, JP21ae0121005, JP21ae0121006, JP21ae0121007, JP20ae0101055, and JP20ae0101060 to A.K.), the CREST program (Grant Number: JPMJCR21N2 to J.I.) and the GteX Program (Grant Number: JPMJGX23B4 to J.I.) from the Japan Science and Technology Agency (JST), and JSPS KAKENHI (Grant Number: JP23K26469 and JP23H01776 to J.I., and JP18K14374 to M.T,) from the Japan Society for Promotion of Science (JSPS).

## Author contributions

M.T., Y.I., M.S., Y.Shoya., N.Hashii., A.K., and J.I. conceptualized and designed the study. M.T., Y.Shima., K.N., Y.I., M.M.-K., K.S., T.M., K.H., and N.Hashiba performed all yeast experiments and analyses. M.S. and Y.Shoya isolated and characterized the IgY antibody #Omi-7. N.Hashii and C.O. performed the LC/MS analysis. M.T., Y.I., M.S., Y.Shoya., N.Hashii., and J.I. wrote the manuscript. All authors agreed to publish the paper.

## Competing interests

M.S. and Y.Shoya are employees of Pharma Foods International Co. Ltd. M.T., K.N., A.K., and J.I. are inventors of pending patent applications submitted by Kobe University that cover the inducible synthetic promoters described in this study (PCT/JP2021/044573, PCT/JP2021/044574, and JP2020202234). Y.I. is the inventor of one of the above pending patent applications (PCT/JP2021/044573). M.T., Y.I., M.M.-K., M.S., Y.Shoya., A.K., and J.I. are inventors of the pending patent application submitted by Kobe University and Pharma Foods International Co. Ltd. that covers the IgY antibody obtained in this study (JP2023188697). The other authors declare no competing interests.

## Additional information

Jun Ishii.

