## [Transparent Peer Review file · Nature Communications]

Designing strong inducible synthetic promoters in yeasts

Corresponding Author: Dr Jun Ishii

Version 0:

Reviewer comments:

Reviewer #1

(Remarks to the Author)

The manuscript presents a very interesting set of results, clearly demonstrating strong levels of induction (and low levels of leakiness) from a set of novel promoters in the yeasts *K. phaffii* (the former *P. pastoris*) and *S. cerevisiae*. The work is of high quality and reported clearly, and will be of particularly strong interest to those working in biological engineering, though it may also find a wider audience among microbiologists more generally.

Questions/comments:

1) Were there any issues encountered in terms of the size of the insulating regions required to achieve the desired induction properties? An addition kilobase (or more) could be a significant amount of additional DNA to add to a construct, particularly in bacterial plasmids but perhaps also in the yeast themselves. If there were actual or potential concerns encountered in terms of the size of the constructs, could you comment on those?

2) For people not involved in large-scale bioproduction, it would be helpful to be more explicit about the motivation for inducible promoters in that context. My understanding is that the idea is to decouple the cell growth and protein production phases by inducing production only after enough biomass has been created, is that correct?

Since one of the key demonstrations provided of the application of these inducible promoters are motivated by bioproduction, some additional commentary on the economics of the system would also be helpful to include. For example, how does the inducer DAPG compare to other inducers (like methanol) in terms of its economic viability? Methanol induction is often used in larger-scale bioproduction, driven in large part (as I understand it) by its relatively low cost, making it practical for use in larger-scale reactors. Is DAPG similarly inexpensive? The general design method could (I assume) be used to create similar promoters to respond to other inducer molecules, but since DAPG is featured so prominently in the results, it would be worthwhile to discuss its features, specifically in the context of larger-scale bioproduction processes.

On a related note, after wondering about these economic issues, I was pleased to see the discussion of the inducer-OFF version of the system, later in the manuscript. That inverted induction logic can address concerns over requiring too much of a potentially expensive molecule ... It might help the reader to note, earlier in the paper, that this version of the promoters will be addressed later. (Or to shift some of the discussion of cost from the inducer-OFF section into an earlier section, so that readers like me who were wondering about that issue can have it addressed sooner.)

3) I had difficulty understanding whether the novel inducible promoters were in fact uniquely required for the work on "difficult to express" proteins, could you comment on this? As currently written, I didn't see how the new promoters were involved, except as an example of a potential protein of interest that could be placed downstream of the promoters? The issue with the SARS-CoV-2 omicron variant appeared to be resolved through modified incubation conditions rather than anything depending on the inducible promoters specifically, is that right? If so, the results still do provide a valid example that the system can be used to carry out bioproduction of an important protein, but perhaps do not specifically illustrate the need for inducible promoters? But I feel like I may not be understanding the current version of the section as written, so comments and clarifications would be helpful.

Reviewer #3

(Remarks to the Author)
General Comments to the authors

This study focused on inducible promoter (iSynP) engineering and characterisation, primarily in the yeast, *Komagataella phaffii* (*Pichia pastoris*). The authors initially developed a model DAPG-inducible promoter and identified leakiness due to a cryptic upstream activating sequence, which was rectified with the introduction of an insulator sequence. Subsequently, a minimal iSynP was developed which was strongly induced in the presence of DAPG, which was reportedly 2-fold stronger than the constitutive KpGAPDH promoter. The authors later maximised induction by increasing the number of operator repeats, which led to a >1,000-fold induction.

The system was later tested for therapeutic protein production, including two nanobodies. Notably, the authors claim a 5-fold higher protein level compared to KpAOX1, which is the gold-standard for protein production in *P. pastoris*. To further demonstrate a potential industrial application (and save costs on expensive reagents), the authors generated an “inducer-OFF” system, in which gene expression is repressed or deactivated in the presence of an inducer. The authors claim a >10-fold higher expression level compared to the KpGAPDH promoter, which is significant. Finally, the authors express viral antigens, immunise chickens with these, and follow up with phage-based biopanning cycles. This demonstrates that the system can produce sufficient amounts of viral antigens.

I would like to congratulate the authors on a well-written and comprehensive body of work. It is clear that extensive effort has been put into this manuscript and the outcomes are significant. Although the methanol-inducible promoter has been used for years, the available existing toolkit of inducible promoters remains limited. The inducer-OFF system is of particular benefit in industry, where the costs (and safety) of inducer reagents is of particular concern.

Despite these comments, I have some concerns and comments which should be addressed.

Ethical Approval on Live Animals

A concern involves the lack of detail on the ethics regarding the use of live animals. No statement on ethics can be found in the main manuscript, nor is there any information on the administering organisation for ethical approval. This is despite the following ticked statements:

--“Ethical compliance: We have complied with all relevant ethical regulations and include a statement affirming this in the manuscript.”

--“Ethics committee: We have disclosed the name(s) of the board and institution that approved the study protocol in the manuscript.”

The only written statement I could find is as follows:

-- Ethics oversight: “All experiments were performed in accordance with the Guidelines for the Care and Use of Laboratory Animals of Pharma Foods International Co. Ltd.”

Further clarification is required regarding the ethical approval of live animals.

General/Major Comments

1. DAPG (2,4-diacetylphloroglucinol) can be toxic to many microbes. Is there any evidence of toxicity or growth inhibition in *P. pastoris*?
2. I notice that the error bars are extremely tight throughout the manuscript (e.g. Figure 3C). The authors claim that the error bars are the means of “at least four independent experiments”. What is meant by an independent experiment? (Noting that an “experimental replicate” has a specific definition, which is separate to technical and biological replicates). I could not find this information in the Materials and Methods section or elsewhere in the manuscript.
3. The study claims a >3-fold and a >10-fold increase in protein levels compared to KpAOX1 and KpGAPDH. These claims are significant. However, information is missing from the Methods section (‘SDS-PAGE analysis’) and it’s worth noting the difference in growth media conditions (e.g. methanol vs glucose). Cell density and stage of growth can have significant effects on promoter activity. What volume of cell culture was used for each protein extraction? Was the optical density normalised for each? Were all cultures and replicates in the same growth stage? This should also be clarified for the other sections in Methods.
4. To follow on from this point, there appears to be a discrepancy in protein levels for DAPG-induced ALX-0171. Is there a reason why levels appear lower in Figure 5B compared to Figure 5A?
5. Regarding the methodology on the use of chickens (page 22), further information is required. Although the manuscript states “total RNA was extracted”, it doesn’t state how this was performed. Was RNA extracted from blood samples, or splenocytes? Were the chickens euthanised?
6. The first component of the research, investigating the addition of insulation sequences to prevent transcriptional activation from upstream cryptic activating sequences was evaluated. However, it was not clear whether this insulator sequence addition was used in the iSynP systems later in the study when studies on the expression of the fluorescent protein and

pharmaceutical proteins were performed, neither for the design of the yeast constructs towards generating viral antigens in vivo. Please clarify.

7. Please clarify the use of statistical significance and associated terminology: Statistical analysis where statements were made about significant differences in fluorescent responses. For example, on page 6: "Replacing a 15-bp sequence downstream of the phlO element with another sequence (Fig. 3a, KpDAS1mut) resulted in a significantly increased fold induction of this promoter". This could be further supported by some basic statistical analyses. Also, in Figure 3b, it may appear that a +70bp spacing will be significantly different from a +40bp, and possibly +80bp spacing in KpAOX1 at least. Statistical analyses will prove that truncating the core promoters downstream will not affect the fold induction of fluorescence, again in strain KpAOX1 at least.

Specific Comments

Please note that the introduction of line numbers would have made reviewing the manuscript significantly easier.

- a) Page 5: The sentence "EGFP expression in the yeast strain was..." should be re-worded for clarity to clarify that auto-fluorescence was a result of leakiness.
- b) Page 5 (Figure 2): What is meant by the following sentence? "Individual and average fluorescence of four independent experiments are indicated by dots and lines, respectively". No averages are shown in the figure.
- c) Page 6: A minor text error – "enhancer." Instead of "enhancer".
- d) Page 7 (Figure 3): The positioning of the letters "A", "B" and "C" on Figure 3 appear out of place. Please correct.
- e) Page 10: I was missing some explanation on how the authors finally decided on a 6X phlO repeat and a 61-bp spacer upstream of the TATA-box sequence. The 6X bacterial operator repeat has some referral looking at the results in Figure 4 and Supplementary Figure 7, but not for the 61-bp spacer selection.
- f) Page 11: The authors fail to describe how the "trifunctional fusion selector" gene works. Although the prior reference was cited, a sentence or two would be of benefit to the reader.
- g) Page 13: Incomplete signal peptide cleavage and nonspecific proteolysis could affect the purity and functionality of the produced proteins. It should be mentioned that further optimization may be necessary to address these issues.
- h) Page 16: The statement on page 16 suggesting that a 94-bp iSynP from the current study will outperform "recently reported strong iSynPs" should be considered with caution in the final manuscript. Although this may be the case, the study did not compare the iSynPs from other studies directly with the newly designed iSynPs from the current study under the same laboratory conditions.
- i) Page 16: In the final paragraph, the study did not produce various pharmaceutical proteins, but rather two pharmaceutical proteins having real pharmacological significance. Consider re-phrasing the sentence.
- j) Page 18 (Methods): Please clarify the subsequent staining method used for visualization of SDS PAGE.
- k) Page 19 (Methods): The two subsections entitled "Inducible production of ALX-0081" and "Selective nanobody production" should be combined, in my opinion. As it currently reads here, both sections start with "For yeast with iSynPs...", making it confusing to follow the different procedures. Rather, it seems that all yeasts with iSynPs were grown overnight in BMDY, from which two routes were taken, either using 5% of the pellet and resuspending it in BMDY+DAPG (except for AOX1 promoter strains). Secondly, a 10% pellet was resuspended in BMDY+DAPG+Dox. Was there a reason for these specific cell pellet percentages? Did the authors test various cell pellet fractions/percentages.
- l) References: The reference list only provides some of the author names. Please list them in full.
- m) Supplementary Figures: In the main text, supporting Figure 4, it is mentioned that 2, 6, 12, and 18 repeats of phlO, were used in the design, but in Supplementary Figure 8b, the illustration shows 7, 14 and 21 repeats (blue "boxes". The same for supplementary Figure 12, showing 7 repeats of phlO, not 6.

Version 1:

Reviewer comments:

Reviewer #1

(Remarks to the Author)

The revisions have addressed all the questions I raised.

Reviewer #3

(Remarks to the Author)
Dear Editor

Thank you for the opportunity to review the revised manuscript of Tominaga et al (2024).

I worked through the revised manuscript and carefully checked whether the authors adequately addressed all of the queries that were raised when I reviewed the original paper.

I am satisfied that the authors positively responded to each of my queries and revised their manuscript accordingly. Therefore, I am happy to recommend that this revised paper be accepted for publication.

Kind regards

Sakkie Pretorius

1. General response

We would like to thank the reviewers for their valuable comments on our manuscript, who particularly emphasized on the following:

1. Feasibility of adding large (>1 kb) insulating sequence (Reviewer #1)
2. Significance of induction system in bioproduction, especially in large-scale, including the economical comparison, conventional, and novel induction systems (Reviewer #1)
3. Ethical statement about the chicken immunization (Reviewer #3)
4. Experimental details including basic statistical analysis, reproducibility, and incubation conditions (Reviewer #3)

We have revised the manuscript following the reviewers' suggestions and major changes are listed below.

Additionally, we thoroughly rechecked the entire manuscript and samples, where we noticed following mistakes that were corrected accordingly. Please note that the corrected data do not change any of the discussion in this article. Other minor mistakes were also corrected. The modifications are highlighted in yellow in the revised manuscript. Now, we are confident that the revised manuscript is free from errors in terms of data, sequence, and sample names.

- **Supplementary Fig. 16b (former Supplementary Fig.14b)**

We noticed the data for 'selected library B' are duplicated and incorrectly plotted as a part of 'selected library A' in Supplementary Fig. 16b (former Supplementary Fig. 14b), that were corrected accordingly.

Original Fig. S16b (former Fig. S14b)

Corrected Fig. S16b

- **Supplementary Fig. 5 (former Supplementary Fig. 4)**

Re-sequencing of the plasmid for DAPG-inducible EGFP expression with randomized upstream vector sequence (appeared in the original **Supplementary Fig. 4**, "N₃₀ variants") revealed that it contains additional mutations in the vector sequence. Therefore, we re-constructed the plasmid with the exact sequence documented in the manuscript and repeated the experiment again (please see below). The related description in the manuscript (page 6, lines 127 and 128) were also revised.

Original Fig. S5 (former Fig. S4)

Corrected Fig. S5

Related description;

‘We identified 69 unique mutant sequences of which 40 sequences reduced iSynP leakiness by >5-fold, indicating that the original 30-bp sequence may act as an “enhancer.”’

2. List of major changes (A-D)

The revised manuscript is shown in blue and the modifications are highlighted in yellow.

- A) As mentioned above, Supplementary Figs. 5 and 16 (former Supplementary Figs. 4 and 14) and the related description in the manuscript (page 6, lines 127 and 128) were revised.
- B) In response to the Reviewer #1 comment, the comparison of induction cost between methanol induction system (*KpAOXI*) and the DAPG-inducible system is added to the Discussion section (page 18, lines 362–367, shown below).

‘Regarding the economic viewpoint, the cost for DAPG induction (10 μ M) is approximately twofold higher than conventional inducer (methanol, 1 [v/v]%). Applying the design principle to construct inducible systems described in this study, other bacterial induction system with cheaper induction molecules, such as choline chloride⁴¹, can be used to construct novel iSynPs to save the induction costs. Alternatively, DAPG-OFF system, which enables the induction by the removal of DAPG, could be used to minimize the induction costs.’

- C) In response to the Reviewer #3 comment, following changes were done:

- a) The description about the “Ethical Approval on Live Animals” was added to the Methods section (pages 25 and 26, lines 573–576). Additionally, detailed information regarding the experiments on chickens was also added (page 24, lines 523–526). The added sentences are as follows.

- **‘Ethics & Inclusion statement**

All animal experiments were performed in accordance with the Guidelines for the Care and Use of Laboratory Animals of Pharma Foods International Co. Ltd. Experimental protocols and studies were reviewed and approved by the Animal Experimentation Management Committee of Pharma Foods International Co., Ltd.’

- ‘Three days after the final immunization, the immunized chickens were euthanized, and the spleens were collected. Total RNA was extracted from the spleens, and cDNA was obtained via reverse transcription using the PolyAtract® mRNA Isolation Systems (Cat. No. Z5310, Promega) and the PrimeScript II 1st Strand cDNA Synthesis Kit (Cat. No. 6210A, Takara Bio).’

- b) “Statistical analysis and reproducibility” section was added and statistical difference (p-value) and correlation coefficient (R) were added where necessary to elaborate each experiment and result including basic statistical analysis (page 25, lines 567–571). Statistical significance is now indicated using asterisks and the legends are revised accordingly. In doing so, we noticed several numerical errors in the **Figs. 3b** and **6b**, and **Supplementary Figs. 12b** (former **Supplementary Figs. 11b**) that were corrected accordingly (shown below).

Statistical analysis and reproducibility

Statistical difference (p-value) was determined with two-tailed t-test. All yeast strains except for *S. cerevisiae* strains harboring centromeric plasmids were stored as glycerol stock and streaked onto YPD plate before experiments. Experiments were performed with the resulting colonies in more than triplicate unless otherwise noted.

• Revised Figure 3.

• Revised Figure 6.

• Revised Supplementary Figure 12.

- c) We added a description about how the trifunctional TBG marker works in yeast during ON and OFF selections and screenings (pages 12, lines 246–255).

‘Briefly, PCR-randomized *phlTA* was cloned into an expression vector in the *S. cerevisiae* strain harboring the iSynP (six copies of *phlO* fused with *ScGAL1* core promoter, [*P_{ScphlO6}*]) followed by the trifunctional fusion selector gene (TB_{D25A}G), encoding herpes simplex virus thymidine kinase, D25A mutant of Zeocin resistance marker (BLE_{D25A}), and monomeric umikino green 1. The resulting yeast library was first subjected to ON selection in the absence of DAPG; yeast variants with strongly activated *P_{ScphlO6}* by the mutant PhlTA could survive under Zeocin selection as BLE_{D25A} inactivates Zeocin (ON selection). Subsequently, in the presence of 5 or 0.5 μM DAPG, yeast variants expressing TB_{D25A}G via the incomplete deactivation of mutant PhlTA by DAPG caused cell death in the presence of 5-Fluoro-2-deoxyuridine because hsvTK convert it to the toxic compound (OFF selection). From the resulting two selected yeast library, a total of 93 clones were ON/OFF screened by measuring in the presence or absence of 5 μM DAPG.’

- d) The description about the “Fluorescence analysis with flow cytometry” and “Small-scale nanobody production” were revised to clarify the yeast cells in the stationary phase were used (page 20, lines 419–429; page 20 and 21, lines 439–448). Besides, we have added the missing information to the latter section to clarify that the mean fluorescence and forward scatter intensity were obtained for the experiments using *Saccharomyces cerevisiae* due to the fluorescence heterogeneity of the cells. We have also revised the Supplementary Fig. 7 to clarify this point.

- **Fluorescence analysis with flow cytometry**

Fluorescence analysis of yeast was performed using a previously described method⁴⁶ with slight modifications. Briefly, *K. phaffii* strains were grown in 500 μL/well YPD media on 96-well plates (Costar[®], #3960 [Corning, Corning, NY, USA]) with different concentrations of inducers (10 μM for DAPG, 30 μg/mL for Dox, and 3 μM for HSL, unless otherwise noted) for 24 h (30°C, 1000 rpm). For KpDAPG-OFF system, 30 μM DAPG was added to the preculture media. The resulting yeast cells in stationary phase were harvested and analyzed using a CytoFLEX (Beckman Coulter, Brea, CA, USA) flow cytometer with a 488-nm laser and a bandpass filter for GFP (525/40 nm). Median fluorescence and forward scatter intensity measurements (i.e., FITC-A and FSC-A) were taken. Cell-size normalized fluorescence was then calculated by dividing the FITC-A signal by the FSC-A signal. Fold induction was calculated by dividing the normalized fluorescence of induced cells by that of uninduced cells. For *S. cerevisiae*, SCSM-His-Leu-Ura (Supplementary Fig. 7) or SD + Met (Supplementary Fig. 16) media were used and mean fluorescence and forward scatter intensity were taken to calculate the cell-size normalized fluorescence.’

- **Small-scale nanobody production**

For the production of ALX-0081 using iSynP or *P_{KpAOXI}*, cells were grown overnight at 30°C and 170 rpm in BMDY (for iSynP) or BMGY (for *P_{KpAOXI}*) media and collected by centrifugation. After washing, 5% of the cell pellet was resuspended in 2 mL of BMDY medium supplemented

with DAPG (10 μ M) for DAPG induction and BMDY medium for methanol induction, and incubated for 60 h. For the selective production of two nanobodies with single yeast, 10% of the cell pellet prepared with the overnight yeast cell culture (30°C, 170 rpm) in BMDY medium was resuspended in 2 mL of BMDY media supplemented with DAPG (10 μ M) and Dox (30 μ g/mL) in different combinations and incubated for 49 h to induce nanobody secretion. After incubation, the cell pellet was harvested via centrifugation, and 50 μ L of the supernatants were then subjected to SDS-PAGE analysis to evaluate nanobody production per volume.

Revised Supplementary Fig. 7 (former Supplementary Fig. 6)

e) Supplementary Figs. 9 and 14 (former Supplementary Figs. 8 and 12) were corrected.

- Corrected Supplementary Fig. 9b (former Supplementary Fig. 8b)

- Corrected Supplementary Fig. 14 (former Supplementary Fig. 12)

3. Point by point responses to the “REVIEWER COMMENTS”

The author’s responses and the revised manuscript are shown in red and blue. The modifications in the revised manuscript are highlighted in yellow.

Reviewer #1 (Remarks to the Author):

The manuscript presents a very interesting set of results, clearly demonstrating strong levels of induction (and low levels of leakiness) from a set of novel promoters in the yeasts *K. phaffi* (the former *P. pastoris*) and *S. cerevisiae*. The work is of high quality and reported clearly, and will be of particularly strong interest to those working in biological engineering, though it may also find a wider audience among microbiologists more generally.

[Author response]

We appreciate for your insightful comments. We have revised the manuscript accordingly.

Questions/comments:

1) Were there any issues encountered in terms of the size of the insulating regions required to achieve the desired induction properties? An addition kilobase (or more) could be a significant amount of additional DNA to add to a construct, particularly in bacterial plasmids but perhaps also in the yeast themselves. If there were actual or potential concerns encountered in terms of the size of the constructs, could you comment on those?

[Author response]

Thank you for your comment. An addition of more than 1-kb insulating sequence to every single expression cassette could be troublesome when multiple expression cassettes are assembled to construct large metabolic pathways or genetic circuits. We have discussed this point in the Discussion section as shown below (page 17, lines 334–337).

‘Regardless of its source, any sequence can potentially act as a weak activator when placed upstream of a TATA-box in an iSynP, leading to unintended basal expression (leakiness). This cryptic activation is difficult to predict, but we have shown that leakiness from iSynPs could be largely mitigated by insulation (Fig. 2), increasing operator repeats (Fig. 4 and Supplementary Fig. 8), and mutating causal operators (Fig. 4c and Supplementary Fig. 12b). Although mutating BOs may inhibit the binding in some cases, the DNA binding of bacterial transcription factors to mutant operators may be improved through domain swapping³⁸ or fast-directed evolution strategies^{13, 39, 40}. By effectively mitigating leakiness, we can achieve strong iSynP construction through the direct fusion of bacterial operator repeats to the TATA-box sequence. Considering all our findings, we established a generalizable design for creating potent iSynPs in various eukaryotic yeasts (Fig. 1). Although large DNA constructs could be more stably assembled in yeasts⁴¹, adding more than 1-kb insulating sequence to upstream of each iSynP is a potential drawback for construction of multigene pathways or complex, higher-order genetic circuits. Comprehensive studies that could identify “sequence-leakiness relationship” could further simplify the iSynP design.’

2) For people not involved in large-scale bioproduction, it would be helpful to be more explicit about the motivation for inducible promoters in that context. My understanding is that the idea is to decouple the cell growth and protein production phases by inducing production only after enough biomass has been created, is that correct?

[Author response]

Thank you for your comment. We agree that decoupling the cell growth and protein production phases is one of the main purposes of using inducible promoters. To clarify this point, we have revised the Introduction section as follows (page 2, lines 36 and 37):

‘Artificial biological activities, such as maximizing metabolite production ¹, producing biotherapeutics ^{2,3}, and reprogramming cellular behavior using synthetic genetic circuits ⁴, may be accomplished by modulating gene expression. Inducible promoters are essential building blocks for this purpose. Specifically, decoupling the cell growth and protein production phases is a major purpose of using inducible promoters ⁵. Their programmability and wide dynamic range allow for precise control of target gene expression. Consequently, identifying or engineering effective inducible promoters is a crucial step in most synthetic biology projects. Because of the lack of thorough understanding of their sequence-function relationships compared to their prokaryotic counterparts, eukaryotic inducible promoters are usually more difficult to engineer ⁶, even in yeast, a well-characterized eukaryotic microbial model. This could be partially attributed to their long promoter sequence containing multiple sequence motifs, in which endogenous factors bind to control promoter activity in an integrated fashion. Therefore, yeast synthetic biology still relies on a set of well-known promoters ⁷.’

Since one of the key demonstrations provided of the application of these inducible promoters are motivated by bioproduction, some additional commentary on the economics of the system would also be helpful to include. For example, how does the inducer DAPG compare to other inducers (like methanol) in terms of its economic viability? Methanol induction is often used in larger-scale bioproduction, driven in large part (as I understand it) by its relatively low cost, making it practical for use in larger-scale reactors. Is DAPG similarly inexpensive? The general design method could (I assume) be used to create similar promoters to respond to other inducer molecules, but since DAPG is featured so prominently in the results, it would be worthwhile to discuss its features, specifically in the context of larger-scale bioproduction processes.

[Author response]

Thank you for your comment. Accordingly, we added the related description to the Discussion section as shown below (page 18, lines 362–367);

‘Regarding the economic viewpoint, the cost for DAPG induction (10 μ M) is approximately twofold higher than conventional inducer methanol (1 [v/v]%). Applying the design principle to construct inducible systems described in this study, other bacterial induction system with cheaper induction molecules, such as choline chloride ⁴⁴, can be used to construct novel iSynPs to save the induction costs. Alternatively, DAPG-OFF system, which enables the induction by the removal of DAPG, could be used to minimize the induction costs.’

On a related note, after wondering about these economic issues, I was pleased to see the discussion of the inducer-OFF version of the system, later in the manuscript. That inverted induction logic can address concerns over requiring too much of a potentially expensive molecule ... It might help the reader to note, earlier in the paper, that this version of the promoters will be addressed later. (Or to shift some of the discussion of cost from the inducer-OFF section into an earlier section, so that readers like me who were wondering about that issue can have it addressed sooner.)

[Author response]

Thank you for your comment. Accordingly, we have added the related description in the last paragraph of the Introduction section (page 3, lines 68–70).

‘In this study, we present a generic design for constructing tightly regulatable iSynPs in yeast (Fig. 1). We found that strong yeast iSynPs can be constructed by (1) inserting >1-kbp insulator sequences to prevent transcriptional activation from upstream cryptic activating sequences, (2) directly fusing operators upstream of the TATA-box, and (3) increasing operator repeats and/or screening (mutating) bacterial operators to reduce their cryptic activation without compromising binding to sTAs. Based on these rules, we constructed a series of inducible expression constructs with >10³-fold induction in reporter gene expression, one of which can be induced by removing the inducer. These induction systems were validated by the production of two different pharmaceutical proteins as well as the cost-effective, inducer-free large-scale overproduction of a single value-added protein with a titer of up to 2 g/L. We also demonstrated the utility of the promoters by producing an omicron variant of SARS-CoV-2 receptor binding domain (RBD), which is notoriously difficult to express in microbes²⁰ and may be readily used for chicken immunization to obtain antigen-specific antibodies. The expression systems described herein may be used in the future to produce various pharmaceutical proteins in a highly flexible manner for several purposes and to reprogram metabolic pathways.’

3) I had difficulty understanding whether the novel inducible promoters were in fact uniquely required for the work on "difficult to express" proteins, could you comment on this? As currently written, I didn't see how the new promoters were involved, except as an example of a potential protein of interest that could be placed downstream of the promoters? The issue with the SARS-CoV-2 omicron variant appeared to be resolved through modified incubation conditions rather than anything depending on the inducible promoters specifically, is that right? If so, the results still do provide a valid example that the system can be used to carry out bioproduction of an important protein, but perhaps do not specifically illustrate the need for inducible promoters? But I feel like I may not be understanding the current version of the section as written, so comments and clarifications would be helpful.

[Author response]

Thank you for your comment. We intended to demonstrate that our novel promoters can be used to produce not only well-engineered nanobodies, but also naturally occurring, practical target like RBD^{om}. As you pointed out, we did not evaluate the need for the novel inducible promoter to produce this protein. To clarify these points, we have revised the related section as shown below (pages 13 and 14, lines 277–282).

‘Production of a difficult-to-express virus antigen for animal immunization

We evaluated whether our expression platform can be applied not only to produce well-engineered nanobodies like ALX-0081, but also to produce “difficult-to-express” proteins, such as virus antigen proteins. Accordingly, we attempted to produce an RBD from a SARS-CoV-2 omicron variant (RBD^{om}) harboring 15 mutations known to reduce stability and expression^{20, 36}, using the DAPG-inducible (KpDAPG-ON) system. The gene encoding RBD^{om} was cloned downstream of the DAPG-iSnyP insulated with 1-kb *ARG4* sequence and integrated into *K. phaffii* strain expressing rPhITA_{2-1E}. The resulting yeast strain produced an RBD^{om} variant when incubated at 18°C rather than 30°C (Fig. 7a) consistent with the reported instability of RBD^{om}^{20, 36}, yielding 2.5-mg purified protein when cultured in 225 mL in test tubes (Supplementary Fig. 19).

Reviewer #3 (Remarks to the Author):

General Comments to the authors

This study focused on inducible promoter (iSynP) engineering and characterisation, primarily in the yeast, *Komagataella phaffii* (*Pichia pastoris*). The authors initially developed a model DAPG-inducible promoter and identified leakiness due to a cryptic upstream activating sequence, which was rectified with the introduction of an insulator sequence. Subsequently, a minimal iSynP was developed which was strongly induced in the presence of DAPG, which was reportedly 2-fold stronger than the constitutive KpGAPDH promoter. The authors later maximised induction by increasing the number of operator repeats, which led to a >1,000-fold induction.

The system was later tested for therapeutic protein production, including two nanobodies. Notably, the authors claim a 5-fold higher protein level compared to KpAOX1, which is the gold-standard for protein production in *P. pastoris*. To further demonstrate a potential industrial application (and save costs on expensive reagents), the authors generated an “inducer-OFF” system, in which gene expression is repressed or deactivated in the presence of an inducer. The authors claim a >10-fold higher expression level compared to the KpGAPDH promoter, which is significant. Finally, the authors express viral antigens, immunise chickens with these, and follow up with phage-based biopanning cycles. This demonstrates that the system can produce sufficient amounts of viral antigens.

I would like to congratulate the authors on a well-written and comprehensive body of work. It is clear that extensive effort has been put into this manuscript and the outcomes are significant. Although the methanol-inducible promoter has been used for years, the available existing toolkit of inducible promoters remains limited. The inducer-OFF system is of particular benefit in industry, where the costs (and safety) of inducer reagents is of particular concern.

[Author response]

We appreciate for your insightful comments. We have revised the manuscript accordingly.

Despite these comments, I have some concerns and comments which should be addressed.

Ethical Approval on Live Animals

A concern involves the lack of detail on the ethics regarding the use of live animals. No statement on ethics can be found in the main manuscript, nor is there any information on the administrating organisation for ethical approval. This is despite the following ticked statements:

--“Ethical compliance: We have complied with all relevant ethical regulations and include a statement affirming this in the manuscript.”

--“Ethics committee: We have disclosed the name(s) of the board and institution that approved the study protocol in the manuscript.”

The only written statement I could find is as follows:

-- Ethics oversight: “All experiments were performed in accordance with the Guidelines for the Care and Use of Laboratory Animals of Pharma Foods International Co. Ltd.”

Further clarification is required regarding the ethical approval of live animals.

[Author response]

Thank you for pointing this out. We have added the ethics statement in the Methods section as shown below (pages 25 and 26, lines 572-575):

Ethics & Inclusion statement

All animal experiments were performed in accordance with the Guidelines for the Care and Use of Laboratory Animals of Pharma Foods International Co. Ltd. Experimental protocols and studies were reviewed and approved by the Animal Experimentation Management Committee of Pharma Foods International Co., Ltd.

General/Major Comments

1. DAPG (2,4-diacetylphloroglucinol) can be toxic to many microbes. Is there any evidence of toxicity or growth inhibition in *P. pastoris*?

[Author response]

Thank you for your comment. We understand the concern about the toxicity of DAPG. We evaluated the cell growth of *P. pastoris* in the presence of 50 μ M DAPG and did not observe any growth inhibition.

Accordingly, we added the new figure (Supplementary Fig. 1) and the description to the main manuscript as shown below (page 5, lines 97–98);

‘The resulting expression cassette was introduced into yeast expressing rPhlTA. EGFP expression in the yeast strain was >100-fold higher than yeast without the plasmid even in the absence of DAPG and was increased 8-fold following the addition of DAPG (Fig. 2b, X = 0). Note that DAPG had no detectable toxicity even at concentration of 50 μ M in *K. phaffi* (Supplementary Fig. 1).’

Supplementary Fig. 1. Cell growth of *K. phaffii* strain expressing rPhlTA_{2-1E} in the presence and absence of 50 μM DAPG. *K. phaffii* strain PpMT146 was grown at 30°C and 170 rpm in 20 mL YPD media with or without 50 μM DAPG (initial OD₆₆₀ = 0.1). OD₆₆₀ was monitored at each time point. Error bars represent the mean ± SD of three independent experiments. ns, not significant.

2. I notice that the error bars are extremely tight throughout the manuscript (e.g. Figure 3C). The authors claim that the error bars are the means of “at least four independent experiments”. What is meant by an independent experiment? (Noting that an “experimental replicate” has a specific definition, which is separate to technical and biological replicates). I could not find this information in the Materials and Methods section or elsewhere in the manuscript.

[Author response]

Thank you for your comment. All the experiments were performed with biological replicates. We added the following description to the “Statistical analysis and reproducibility” section to clarify this point (page 25, lines 567–571).

Statistical analysis and reproducibility

Statistical difference (p-value) was determined with two-tailed t-test. All yeast strains except for *S. cerevisiae* strains harboring centromeric plasmids were stored as glycerol stock and streaked onto YPD plate before experiments. Experiments were performed with the resulting colonies in more than triplicate unless otherwise noted.

3. The study claims a >3-fold and a >10-fold increase in protein levels compared to KpAOX1 and KpGAPDH. These claims are significant. However, information is missing from the Methods section (‘SDS-PAGE analysis’) and it’s worth noting the difference in growth media conditions (e.g. methanol vs glucose). Cell density and stage of growth can have significant effects on promoter activity. What volume of cell culture was used for each protein extraction? Was the optical density normalised for each? Were all cultures and replicates in the same growth stage? This should also be clarified for the other sections in Methods.

[Author response]

Thank you for your comment. For small-scale nanobody secretion, the supernatant of the stationary phase cultures was analyzed to evaluate protein production per volume (without normalization to optical cell density). For the fluorescent measurement, cells in the stationary phase were analyzed. Note that mean fluorescence and forward scatter intensity were obtained for the experiments using *Saccharomyces cerevisiae* due to the fluorescence heterogeneity of the cells. We have revised the Methods section and Supplementary Fig. 7 to clarify this point.

Small-scale nanobody production

For the production of ALX-0081 using iSynP or P_{KpAOXI} , cells were grown overnight at 30°C and 170 rpm in BMDY (for iSynP) or BMGY (for P_{KpAOXI}) media and collected by centrifugation. After washing, 5% of the cell pellet was resuspended in 2 mL of BMDY medium supplemented with DAPG (10 µM) for DAPG induction and BMMY medium for methanol induction, and incubated for 60 h. For the selective production of two nanobodies with single yeast, 10% of the cell pellet prepared with the overnight yeast cell culture (30°C, 170 rpm) in BMDY medium was resuspended in 2 mL of BMDY media supplemented with DAPG (10 µM) and Dox (30 µg/mL) in different combinations and incubated for 49 h to induce nanobody secretion. After incubation, the cell pellet was harvested via centrifugation, and 50 µL of the supernatants were then subjected to SDS-PAGE analysis to evaluate nanobody production per volume.

Fluorescence analysis with flow cytometry

Fluorescence analysis of yeast was performed using a previously described method⁴⁹ with slight modifications. Briefly, *K. phaffii* strains were grown in 500 µL/well YPD media on 96-well plates (Costar®, #3960 [Corning, Corning, NY, USA]) with different concentrations of inducers (10 µM for DAPG, 30 µg/mL for Dox, and 3 µM for HSL, unless otherwise noted) for 24 h (30°C, 1000 rpm). For KpDAPG-OFF system, 30 µM DAPG was added to the preculture media. The resulting yeast cells in stationary phase were harvested and analyzed using a CytoFLEX (Beckman Coulter, Brea, CA, USA) flow cytometer with a 488-nm laser and a bandpass filter for GFP (525/40 nm). Median fluorescence and forward scatter intensity measurements (i.e., FITC-A and FSC-A) were taken. Cell-size normalized fluorescence was then calculated by dividing the FITC-A signal by the FSC-A signal. Fold induction was calculated by dividing the normalized fluorescence of induced cells by that of uninduced cells. For *S. cerevisiae*, SCSM-His-Leu-Ura (Supplementary Fig. 7) or SD + Met (Supplementary Fig. 16) media were used and mean fluorescence and forward scatter intensity were taken to calculate the cell-size normalized fluorescence.

‘Supplementary Fig. 7. Construction of minimal iSynPs in *S. cerevisiae*. **a.** Schematic illustration of DAPG-iSynPs for *S. cerevisiae* with different spacer lengths between the TATA-box and the start codon. **b.** DAPG-induced GFP expression of the *S. cerevisiae* strains harboring the DAPG-iSynPs were measured Flow-cytometry. **c.** Fold-induction was calculated by dividing GFP fluorescence in the presence of 3 μM DAPG by the uninduced GFP fluorescence in its absence, as measured by flow cytometry. mUkG1, monomeric variant of umikinoko-green 1; N, NheI; S, Sall. TATA, TATA-box. Error bars represent the mean \pm SD of four independent experiments. The double asterisk represents $p < 0.01$. ns, not significant.’

4. To follow on from this point, there appears to be a discrepancy in protein levels for DAPG-induced ALX-0171. Is there a reason why levels appear lower in Figure 5B compared to Figure 5A?

[Author response]

Thank you for your comment. We speculate that this may be due to the expression levels of rPhITA_{2-1E} which were different between these experiments. To enable the simultaneous expression of rPhITA_{2-1E} and rTetTA_{K8N}, expression cassette for rPhITA_{2-1E} is needed to be integrated into the different locus. This might result in a reduced expression of rPhITA_{2-1E}. Moreover, simultaneous expression of two synthetic transcription activators (sTAs) would reduce the expression levels of both sTAs; however, the effect would be more critical for the larger nanobody, ALX-0171.

5. Regarding the methodology on the use of chickens (page 22), further information is required. Although the manuscript states “total RNA was extracted”, it doesn’t state how this was performed. Was RNA extracted from blood samples, or splenocytes? Were the chickens euthanised?

[Author response]

Thank you for your comment. We added the information to the related section as follows (page 24 lines 523–526);

‘Three days after the final immunization, the immunized chickens were euthanized, and the spleens were collected. Total RNA was extracted from the spleens, and cDNA was obtained via reverse transcription using the PolyAtract® mRNA Isolation Systems (Cat. No. Z5310, Promega) and the PrimeScript II 1st Strand cDNA Synthesis Kit (Cat. No. 6210A, Takara Bio).

6. The first component of the research, investigating the addition of insulation sequences to prevent transcriptional activation from upstream cryptic activating sequences was evaluated. However, it was not clear whether this insulator sequence addition was used in the iSynP systems later in the study when studies on the expression of the fluorescent protein and pharmaceutical proteins were performed, neither for the design of the yeast constructs towards generating viral antigens in vivo. Please clarify.

[Author response]

Thank you for your valuable comment. To clarify this point, we added the following description at the end of the 1st section (page 6, lines 131 and 132).

‘In all of the subsequent experiments using *K. phaffii*, 1.6-kb *ARG4* sequence was placed upstream of iSynPs.’

7. Please clarify the use of statistical significance and associated terminology: Statistical analysis where statements were made about significant differences in fluorescent responses. For example, on page 6: “Replacing a 15-bp sequence downstream of the phlO element with another sequence (Fig. 3a, KpDAS1mut) resulted in a significantly increased fold induction of this promoter”. This could be further supported by some basic statistical analyses. Also, in Figure 3b, it may appear that a +70bp spacing will be significantly different from a +40bp, and possibly +80bp spacing in KpAOX1 at least. Statistical analyses will prove that truncating the core promoters downstream will not affect the fold induction of fluorescence, again in strain KpAOX1 at least.

[Author response]

Thank you for your comment. We added the “Statistical analysis and reproducibility” section in the Methods section shown below (page 25, lines 567-571), and the statistical significance was defined as p-value calculated via the parametric t-test (significance was set at 0.05). For instance, we revised the Figure 3 as follows:

‘Statistical analysis and reproducibility

Statistical difference (p-value) was determined with two-tailed t-test. All yeast strains except for *S. cerevisiae* strains harboring centromeric plasmids were stored as glycerol stock and streaked onto YPD plate before experiments. Experiments were performed with the resulting colonies in more than triplicate unless otherwise noted.’

Fig. 3. Optimizing design constraints for strong minimal iSynPs in yeast. **a.** Schematic illustration of DAPG-iSynPs with different spacer lengths between *phiO* element and the TATA-box (X [bp]) and between the TATA-box and the start codon (Y [bp]). **b,c.** Effect of the sequence length between the *phiO* element and the TATA-box (**b**) and between the TATA-box and the start codon (**c**) on induced gene expression driven by the DAPG-iSynPs. Fold induction was calculated by dividing EGFP fluorescence in the presence of 10 μM DAPG by the uninduced EGFP fluorescence in its absence, as measured by flow cytometry. **d,e.** Comparison of induced gene expression level between the conventional *KpGAPDH* promoter system ($P_{KpGAPDH}$) and two iSynPs: a 94-bp iSynP with a 53-bp *KpAOX1* core promoter and an 83-bp iSynP with a 47-bp *DAS1* core promoter. Schematic comparison of these promoters is shown in **d**. Error bars represent the mean ± SD of at least four independent experiments. The single and double asterisk represents $p < 0.05$ and $p < 0.01$, respectively. The promoter architecture of these iSynPs is shown in Supplementary Fig. 6. DAPG, 2,4-diacetylphloroglucinol; EGFP, enhanced green fluorescent protein; iSynP, inducible synthetic promoter; ns, not significant; TATA, TATA-box; WT, wild-type.

Specific Comments

Please note that the introduction of line numbers would have made reviewing the manuscript significantly easier.

[Author response]

We would like to apologize for this inconvenience. The line numbers were added to the revised manuscript.

a) Page 5: The sentence “EGFP expression in the yeast strain was...” should be re-worded for clarity to clarify that auto-fluorescence was a result of leakiness.

[Author response]

Thank you for your valuable comment. We have revised the sentence as shown below (page 5, lines 95–97).

EGFP expression in the yeast strain was >100-fold higher than yeast without the plasmid even in the absence of DAPG and was increased 8-fold following the addition of DAPG (Fig. 2b, X = 0).

b) Page 5 (Figure 2): What is meant by the following sentence? “Individual and average fluorescence of four independent experiments are indicated by dots and lines, respectively”. No averages are shown in the figure.

[Author response]

Thank you for pointing this out. We revised Fig. 2 and the description as follows (page 6, lines 112 and 113);

Error bars represent the mean ± SD of four independent experiments.

c) Page 6: A minor text error – “enhancer.” Instead of “enhancer”.

[Author response]

Thank you for pointing this out. We have revised the sentence accordingly;

d) Page 7 (Figure 3): The positioning of the letters “A”, “B” and “C” on Figure 3 appear out of place. Please correct.

[Author response]

Thank you for pointing this out. We revised Fig. 3 accordingly as shown below.

e) Page 10: I was missing some explanation on how the authors finally decided on a 6X phI0 repeat and a 61-bp spacer upstream of the TATA-box sequence. The 6X bacterial operator repeat has some referral looking at the results in Figure 4 and Supplementary Figure 7, but not for the 61-bp spacer selection.

[Author response]

Thank you for your comment. We added the description of this promoter (page 10, lines 219 and 220) and the new data (Supplementary Fig. 13) for the induction performance of this promoter as shown below:

‘Yeast-based biotherapeutic production platform based on the designed iSynPs

To demonstrate the capability of the promoters for inducing protein secretion, the KpDAPG-ON system was used to produce various pharmaceutical proteins, including nanobody® ALX-0171 (gontivimab), a potent treatment for respiratory syncytial virus infection ³⁰, and nanobody® ALX-0081 (also known as caplacizumab or Cablivi™), which is used to treat acquired thrombotic thrombocytopenic purpura ³¹. Genes encoding these nanobodies were C-terminally fused to the mutant secretion signal sequence, MFα (2×Adv) ³², and cloned downstream of the 419-bp strong DAPG-iSynP, $P_{KpPhl6.0}$, containing a 6 *phlO* repeat a 61-bp spacer upstream of the TATA-box, one of the DAPG-iSynP that could be induced more than 1,500-fold upon the addition of 10 μM DAPG (Fig. 5a and Supplementary Figs. 13 and 14). In these strains, each protein was successfully produced only in the presence of DAPG, with titers >5-fold higher than those obtained using a methanol induction system ($KpAOX1$ promoter).’

Supplementary Fig. 13. Switching performance of DAPG-ON system using iSynPs with different spacer length. a. Schematic illustration of the strain with KpDAPG-ON system, in which rPhlTA_{2-1E} binds to $P_{KpPhl6.0}$ and $P_{KpPhl6.1}$ with 61- and 11-bp spacer sequence between *phlO* repeats and TATA-box to activate EGFP expression. **b.** Dose-response curves for the two KpDAPG-ON systems. The double asterisk represents $p < 0.01$.

f) Page 11: The authors fail to describe how the “trifunctional fusion selector” gene works. Although the prior reference was cited, a sentence or two would be of benefit to the reader.

[Author response]

Thank you for your comment. We added the explanation as follows (page 12, lines 246–255):

‘Briefly, PCR-randomized *phlTA* was cloned into an expression vector in the *S. cerevisiae* strain harboring the iSynP (six copies of *phlO* fused with *ScGAL1* core promoter, [$P_{ScphlO6}$]) followed by the trifunctional fusion selector gene (TB_{D25A}G), encoding herpes simplex virus thymidine kinase, D25A mutant of Zeocin resistance marker (BLE_{D25A}), and monomeric umikino green 1 (mUkG1). The resulting yeast library was first subjected to ON selection in the absence of DAPG; yeast

variants with strongly activated *P_{ScephO6}* by the mutant PhlTA could survive under Zeocin selection as BLE_{D25A} inactivates Zeocin (ON selection). Subsequently, in the presence of 5 or 0.5 μ M DAPG, yeast variants expressing TB_{D25A}G *via* the incomplete deactivation of mutant PhlTA by DAPG caused cell death in the presence of 5-Fluoro-2-deoxyuridine because hsvTK convert it to the toxic compound (OFF selection). From the resulting two selected yeast library, a total of 93 clones were ON/OFF screened by measuring in the presence or absence of 5 μ M DAPG.’

g) Page 13: Incomplete signal peptide cleavage and nonspecific proteolysis could affect the purity and functionality of the produced proteins. It should be mentioned that further optimization may be necessary to address these issues.

[Author response]

Thank you for your valuable comment. We added the following sentence to the last paragraph of the Discussion section (page 17, lines 355–357).

Although the incomplete signal peptide cleavage and the nonspecific proteolysis during the storage period remain to be addressed for future studies, the former may be alleviated by inserting appropriate linker⁴³.

h) Page 16: The statement on page 16 suggesting that a 94-bp iSynP from the current study will outperform “recently reported strong iSynPs” should be considered with caution in the final manuscript. Although this may be the case, the study did not compare the iSynPs from other studies directly with the newly designed iSynPs from the current study under the same laboratory conditions.

[Author response]

Thank you for your valuable comment. We added the following sentence to the related sentence highlighted in yellow (page 17, lines 343 and 344).

Surprisingly, we found that only a 94-bp sequence could work as a promoter to induce downstream reporter expression $>10^3$ -fold, which outperforms recently reported strong yeast iSynPs^{1, 4, 13, 42}, although they might not be compared directly.

i) Page 16: In the final paragraph, the study did not produce various pharmaceutical proteins, but rather two pharmaceutical proteins having real pharmacological significance. Consider re-phrasing the sentence.

[Author response]

Thank you for pointing this out. We have revised the sentence as follows (page 17, line 350):

The constructed iSynPs were used to produce two pharmaceutical proteins (Figs. 5 and 6).

j) Page 18 (Methods): Please clarify the subsequent staining method used for visualization of SDS PAGE.

[Author response]

Thank you for pointing this out. We have revised the sentence as follows (page 20, lines 436 and 437):

SDS-PAGE analysis

An equivalent of 50 μ L of the supernatant obtained by centrifugation of each yeast cell culture was mixed with 10 μ L of 6 \times sample buffer containing a reducing reagent (Cat. No. 09499-14, Nacalai Tesque). This was followed by incubation at $>95^{\circ}\text{C}$ for more than 10 min. Next, 15 μ L of the resultant lysate was separated on a 15% polyacrylamide gel using SDS-PAGE (e-PAGEL [Cat. No. E-R15L, ATTO, Tokyo, Japan]) or 5%–20% gradient polyacrylamide gel (m-PAGEL [Cat. No. M-520L, ATTO]) using SDS-PAGE and stained with CBB Stain One (Nacalai Tesque).

k) Page 19 (Methods): The two subsections entitled “Inducible production of ALX-0081” and “Selective nanobody production” should be combined, in my opinion. As it currently reads here, both sections start with “For yeast with iSynPs...”, making it confusing to follow the different procedures. Rather, it seems that all yeasts with iSynPs were grown overnight in BMDY, from which two routes were taken, either using 5% of the pellet and resuspending it in BMDY+DAPG (except for AOX1 promoter strains).

[Author response]

Thank you for your valuable comment. We have combined the two sections as shown below (pages 20 and 21, lines 439–448):

Small-scale nanobody production

For the production of ALX-0081 using iSynP or P_{KpAOX1} , cells were grown overnight at 30°C and 170 rpm in BMDY (for iSynP) or BMGY (for P_{KpAOX1}) media and collected by centrifugation. After washing, 5% of the cell pellet was resuspended in 2 mL of BMDY medium supplemented with DAPG (10 μM) for DAPG induction and BMMY medium for methanol induction, and incubated for 60 h. For the selective production of two nanobodies with single yeast, 10% of the cell pellet prepared with the overnight yeast cell culture (30°C , 170 rpm) in BMDY medium was resuspended in 2 mL of BMDY media supplemented with DAPG (10 μM) and Dox (30 $\mu\text{g}/\text{mL}$) in different combinations and incubated for 49 h to induce nanobody secretion. After incubation, the cell pellet was harvested *via* centrifugation, and 50 μ L of the supernatants were then subjected to SDS-PAGE analysis to evaluate nanobody production per volume.

Secondly, a 10% pellet was resuspended in BMDY+DAPG+Dox. Was there a reason for these specific cell pellet percentages? Did the authors test various cell pellet fractions/percentages.

[Author response]

Thank you for your comment. We only tested the experimental conditions described in the manuscript. As you pointed out, screening of suitable cell pellet fractions/percentages is an important factor for protein production and we think that this would be addressed in future studies.

l) References: The reference list only provides some of the author names. Please list them in full.

[Author response]

Thank you for pointing this out. We have revised the reference list accordingly.

m) Supplementary Figures: In the main text, supporting Figure 4, it is mentioned that 2, 6, 12, and 18 repeats of *phlO*, were used in the design, but in Supplementary Figure 8b, the illustration shows 7, 14 and 21 repeats (blue “boxes”. The same for supplementary Figure 12, showing 7 repeats of *phlO*, not 6.

[Author response]

Thank you for pointing this out. We have revised the Supplementary Figs. 9b and 14 (former Supplementary Figs. 8b and 14) accordingly as shown below.

b

Corrected Supplementary Fig. 9b (former Supplementary Fig. 8b)

Corrected Supplementary Fig. 14 (former Supplementary Fig. 12)

Author response

We thank all the reviewers for their thoughtful comments. We have revised all related documents to comply with the journal policies. All the corresponding changes in main text file are made with the track-changes in the revised manuscript. Any track changes in supplementary information files are removed.

Reviewer #1 (Remarks to the Author):

The revisions have addressed all the questions I raised.

Response

We appreciate the reviewer's help to improve our manuscript.

Reviewer #3 (Remarks to the Author):

Thank you for the opportunity to review the revised manuscript of Tominaga et al (2024).

I worked through the revised manuscript and carefully checked whether the authors adequately addressed all of the queries that were raised when I reviewed the original paper.

I am satisfied that the authors positively responded to each of my queries and revised their manuscript accordingly. Therefore, I am happy to recommend that this revised paper be accepted for publication.

Response

We appreciate the reviewer's help to improve our manuscript.